# Phloroglucinol α-Pyrones from *Helichrysum*: A Review on Structural Diversity, Plant Distribution and Isolation

**DOI:** 10.3390/plants14223460

**Published:** 2025-11-12

**Authors:** Yulian Voynikov

**Affiliations:** Department of Chemistry, Faculty of Pharmacy, Medical University, 1000 Sofia, Bulgaria; y_voynikov@pharmfac.mu-sofia.bg

**Keywords:** *Helichrysum*, phloroglucinol, α-pyrone, arzanol, isolation, biological activity

## Abstract

*Helichrysum* species (Asteraceae) are renowned for their diverse phytochemical profiles and traditional medicinal applications. Among their specialized metabolites, phloroglucinol-α-pyrone derivatives represent a structurally unique and pharmacologically significant class of compounds. This review consolidates over five decades of phytochemical research, documenting 52 distinct compounds isolated from 11 *Helichrysum* species across the Mediterranean, African, and Iranian regions. The compounds are organized into structural subclasses, including monopyrones, dipyrones, and various phloroglucinol derivatives distinguished by their molecular scaffolds. Isolation yields reported in the literature range from trace amounts to relatively abundant constituents (0.48% *w*/*w*), with arzanol emerging as the most extensively studied compound. Bioactivity profiles reveal anti-inflammatory, antimicrobial, antioxidant, and antiparasitic properties, with arzanol demonstrating potent dual inhibition of mPGES-1 and 5-LOX. This review provides comprehensive reference data for future investigations into the chemistry and therapeutic potential of α-pyrone secondary metabolites from *Helichrysum* species.

## 1. Introduction

The genus *Helichrysum* Mill. (Asteraceae), comprising over 600 species distributed across Africa, Madagascar, the Mediterranean, and Asia, represents one of the most chemically diverse genera within the Asteraceae family. These plants, commonly known as “everlasting flowers” or “immortelle,” have been utilized in traditional medicine for centuries, particularly in Mediterranean and African cultures, for treating various ailments including respiratory disorders, wounds, infections, and inflammatory conditions [1]. The therapeutic properties attributed to *Helichrysum* species have prompted extensive phytochemical investigations over the past decade, revealing a rich and structurally diverse array of secondary metabolites [2,3,4].

Among the various classes of compounds isolated from *Helichrysum* species, phloroglucinol derivatives conjugated with α-pyrone are characterized by their unique biosynthetic origin involving the coupling of polyketide-derived phloroglucinol units with α-pyrone rings, and they display remarkable structural diversity (Appendix A). The first reports of these compounds date back to the 1970s, with the pioneering work of Vrkcoč et al., 1975 [5] on *H. arenarium*, followed by extensive investigations by Bohlmann et al., 1980 [6] and Hänsel et al., 1980 [7] on various African and European *Helichrysum* species.

Despite the extensive phytochemical investigations conducted to date, a comprehensive systematic review of phloroglucinol-α-pyrone derivatives from *Helichrysum* species has not been undertaken. The scattered scientific literature, spanning multiple decades, necessitates a consolidated overview that can serve as a reference for researchers in natural products chemistry. Furthermore, understanding the distribution patterns, structural diversity, and quantitative occurrence of these compounds is essential for quality control of *Helichrysum*-based products, chemotaxonomic studies, and the rational selection of species for bioprospecting efforts.

This review presents a comprehensive analysis of reported phloroglucinol-α-pyrone derivatives isolated from Helichrysum species, organized according to their structural features. The analysis covered peer-reviewed literature from the earliest report of isolated substances of this class in 1970 to present (late 2025) through structured database searches (Google Scholar, PubMed) with defined inclusion/exclusion criteria. The compounds were categorized into distinct structural classes: monopyrones, dipyrones, 3-non-alkylated phloroglucinols (3-NA PGs), 3-prenyl- (3-prenyl PGs), 3-geranyl- (3-geranyl PGs), 2-prenyl- (2-prenyl PGs), and 3-prenyl-methoxy phloroglucinols (3-prenyl-methoxy PGs), amino derivatives, benzofuranes, benzopyranes, chromanes, hetero-trimers, and spiroketals. By consolidating isolation yields, geographical distributions, and analytical methodologies, this work aims to provide a valuable resource for future investigations into the chemistry and therapeutic potential of α-pyrone phloroglucinol secondary metabolites from *Helichrysum*.

## 2. Literature Search Methodology

A thorough scientific literature search was conducted to identify all published research articles detailing the isolation, structural characterization, and biological activity of phloroglucinol-α-pyrone derivatives from *Helichrysum* species, which date from 1970 to 2019. The primary electronic databases utilized for the search were Google Scholar and PubMed. The following combination of keywords and search terms were used, such as “*Helichrysum*”, “alpha-pyrone”, “α-pyrone”, and “phloroglucinol”. Search results were subsequently manually screened. Primary research articles detailing the isolation of phloroglucinol-α-pyrone derivatives from any *Helichrysum* species were included, alongside articles reporting the biological activity of these compounds. The review focused exclusively on full-text publications containing isolation and characterization data, as well as data for biological activity. Conversely, review articles (unless used for initial identification of key primary sources), patents, or studies that exclusively focused on essential oils, general *Helichrysum* extracts, or compounds from other metabolite classes were excluded. The specific data points recorded for each compound included its chemical structure, structural subclass, the source *Helichrysum* species, the geographical location of the gathered plant material, the specific plant part used for isolation, the documented isolation yields, and the relevant analytical methodologies used for structure confirmation. Also, all reported biological activities in the literature were documented.

## 3. Phloroglucinol-α-Pyrones from Helichrysum

The structural complexity of *Helichrysum* phloroglucinol-α-pyrone derivatives ranges from simple monopyrones to elaborate heterotrimeric structures (Table 1 and Appendix A).

### 3.1. Alpha-Pyrone Ring Modifications

Across all subclasses, except spiroketals (see Section 16), the α-pyrone ring is uniformly C5-methylated. Its structural variation among different compounds is determined by the C6 alkyl substitution, which can include a range of linear or branched moieties (e.g., methyl, ethyl, propyl, isopropyl, etc.), predominantly methyl or ethyl. In non-monomeric structures, the α-pyrone ring is characteristically linked to the phloroglucinol via a methylene bridge at C3 (Figure 1). In spiroketals, the trihydroxyacylphenone-fused benzofuran is spiro linked to a 2-furan-3-one, instead of an α-pyrone, making them a structurally distinct subclass. In hetero-trimeric PGs (see Section 15), one α-pyrone moiety is linked via a methylene bridge to the phloroglucinol C3, and another at C5, forming a trimeric structure.

### 3.2. Phloroglucinol Ring Modifications

The phloroglucinol core is acylated at C1, forming a 2,4,6-trihydroxyacylphenone scaffold. Across most subclasses, the hydroxyl groups are positioned at C2, C4, and C6. However, 2-prenyl PGs (see Section 10) represent an exception, featuring a 3,5,6-trihydroxyacetophenone framework with the C4 position linked to the α-pyrone ring. The hydroxyl groups typically remain unesterified, though notable exceptions exist. *O*-methylation at C4 is observed in 3-NA PGs (see Section 6) and 3-prenyl methoxy PGs (see Section 8). Amino PGs display *O*-methylation at C6 and uniquely feature an amino substituent at C4—a characteristic not found in other subclasses (see Section 11). Hydroxyl esterification can also result from intramolecular cyclization of prenyl/geranyl side chains at C3 with either the C2 or C4 hydroxyl group, forming benzofuran (see Section 13) or 2H-benzopyran/chromene ring systems (see Section 12 and Section 14).

### 3.3. C3 Side Chain

Both 3-NA PGs and amino PGs lack C3 substitution on the phloroglucinol ring. In rare instances, the C3 prenyl side chain may be hydroxylated, as exemplified by helipyrone (compound **17**). Hetero-trimeric PGs constitute a distinct structural subclass. Instead of a prenyl or geranyl side chain at C3, these compounds feature an additional α-pyrone ring connected to the phloroglucinol core via a methylene bridge, generating a hetero-trimeric structure.

### 3.4. C1 Acyl Group Diversity

Structural diversity within the phloroglucinol ring largely stems from variation in the C1 acyl moiety. Similar to alkylation patterns observed at C6 of the α-pyrone ring, the C1 position accommodates linear or branched alkyl chains ranging from acetyl group to elongated substituents such as propionyl, butyryl, isobutyryl, etc., where acetyl and propionyl are predominant. Notably, only amino PGs exhibit amination of the C1 acyl group.

Table 1 provides a comprehensive summary of the major pyrone and phloroglucinol-type compounds that have been isolated from various *Helichrysum* species. A notable example is arzanol, which has garnered significant attention due to its potent anti-inflammatory, antioxidant, antimicrobial, and neurobehavioural activities [8,9,10]. The isolation of structurally unique compounds such as the helispiroketals (see Section 16) from *H. oocephalum* [11] and the amino-containing helichrytalicines (see Section 11) from *H. italicum* [12] continues to expand our understanding of the biosynthetic capabilities within this genus.

The isolation yields reported in the literature of these compounds range from relatively abundant constituents such as arzanol (up to 0.48% *w*/*w* from *H. stoechas*) [13] to trace quantities like helipyrone B (0.00001% *w*/*w* from *H. oocephalum*) [11]. This variation in yields reflects not only the differential accumulation patterns across species and plant organs but also highlights the analytical challenges associated with the isolation and characterization of minor metabolites from complex plant matrices. Recent advances in analytical techniques, particularly the application of high-resolution mass spectrometry and advanced NMR spectroscopy, have facilitated the discovery of previously undetected minor constituents and enabled more comprehensive metabolite profiling [11,14].

In the sections that follow, we present a detailed account of the major phloroglucinol-α-pyrone derivatives isolated from *Helichrysum* species. For every compound subclass, a comprehensive summary of all documented plant sources, including taxonomic identification, collection locations, and the precise plant organs utilized for extraction, are documented. Methodological details covering extraction solvents, purification protocols, and isolation strategies are reported alongside all available quantitative data (Table 2). The structural features of each subclass, together with representative molecular structures, are described. Also, biological activities are discussed with reference to experimental models and key findings, where available. Appendix A presents molecular formulae, exact masses, structures, SMILES notations, structural subclasses, and corresponding references.plants-14-03460-t001_Table 1Table 1Distribution of phloroglucinol derivatives and related compounds in *Helichrysum* species.Plant NameCompounds DiscoveredPlant PartsRef.***Helichrysum*** ***arenarium* (L.) Moench****Dipyrones**: 


Helipyrone A–C (**4**–**6**)Roots[5]
**2-prenyl PGs**: 


Arenol A (**29**)Aerial parts with flowers[15]***Helichrysum*** ***auriceps* Hilliard****3-prenyl methoxy PGs**: 


Auricepyrone (**19**), 23-Methylauricepyrone (**20**)Roots[6]***Helichrysum*** ***cephaloideum* DC.****3-NA PGs**:


Norauricepyrone (**8**), Methyl-norauricepyrone (**9**)Roots[16]
**3-prenyl methoxy PGs**:


Auricepyrone (**19**)Roots[6,7]
Auricepyrone (**19**)Aerial parts[6]
23-Methylauricepyrone (**20**)Roots[6,7]
23-Methylauricepyrone (**20**)Aerial parts[6]
23-ethyl-6-O-desmethyl-4-O-methylauricepyrone (**21**)Roots[16]
**Benzofuranes**:


22-Methyl-22-ethyl-italipyrone (**37**)Roots[7,16]
22-Methyl-22-propyl-italipyrone (**38**)Aerial parts[7,16]***Helichrysum*** ***italicum* ssp. *microphyllum*****Monopyrones**:


Micropyrone (**3**)Aerial parts (non-woody) with flowers[8,9,17]
**Dipyrones**:


Helipyrone A (**6**)Aerial parts (non-woody) with flowers[8,9,17]
**3-prenyl PGs**:


Arzanol (**10**)Aerial parts (non-woody) with flowers[8,9,17]
Heliarzanol (**17**)Aerial parts (non-woody) with flowers[9]***Helichrysum*** ***italicum* (Roth) G. Don****Monopyrones**:


Micropyrone (**3**)Aerial parts (non-woody) with flowers[14]
**Dipyrones**:


Helipyrone A (**6**)Aerial parts with flowers[7,12,18]
Helitalone A (**7**)non-woody Aerial parts with flowers[14]
**3-prenyl PGs**: 


Arzanol (**10**), Helitalone B (**12**)non-woody Aerial parts with flowers[14]
**Amino PGs**:


Helichrytalicine B (**31**), Helichrytalicine A (**32**)leaf material [12]
**Benzofuranes**:


Italipyrone (**36**)Aerial parts with flowers [7]
**Hetero-trimer PGs**:


Italidipyrone (**43**), 23-Methyl-italidipyrone (**44**)Aerial parts with flowers[7]***Helichrysum*** ***mixtum* (Kuntze) Moeser****3-prenyl PGs**:


23-Methyl-6-O-desmethylauricepyrone (**16**), 23-ethyl-6-O-desmethyl-auricepyrone (**18**)Roots[16]
**Benzopyranes**:
[16]
Isobutyryl-helichromenopyrone (**34**)Roots[16]
Methylbutyryl-helichromenopyrone (**35**)Aerial parts[16]
22-Methyl-22-ethyl-italipyrone (**37**), 22-Methyl-22-propyl-italipyrone (**38**)Roots[16]***Helichrysum*** ***odoratissimum* Sweet.****3-prenyl PGs**:


6-O-Desmethylauricepyrone (**13**), 23-Methyl-6-O-desmethylauricepyrone (**16**)Aerial parts[7]***Helichrysum*** ***oocephalum* Boiss.****Dipyrones**:


Helipyrone A–C (**4–6**)Aerial parts[11]
**3-prenyl PGs**:


Arenol B (**11**), Arenol C (**14**), 3-prenyl norauricepyrone (**15**), 23-Methyl-6-O-desmethylauricepyrone (**16**)Aerial parts[11]
Achyroclinopyrone A (**26**, **28**, **25**, **27**)Aerial parts[11]
**Benzofuranes**:


Helicyclol (**40**)Aerial parts[11]
**Chromane PGs**:


Cycloarzanol C (**41**), Helicepyrone (**42**)Aerial parts[11]
**Hetero-trimer PGs**:


Italidipyrone (**43**), 23-Methyl-italidipyrone (**44**)Aerial parts[11]
**Spiroketals**:


Helispiroketals A-H (**45**, **46**, **49**, **50**, **48**, **47**, **52**, **51**)Aerial parts[11]***Helichrysum*** ***plicatum* DC. ssp. *plicatum*****Benzopyranes**:


Plicatipyrone (**33**)Flower heads[7]***Helichrysum*** ***stenopterum* DC.****3-prenyl PGs**:


23-Methyl-6-O-desmethylauricepyrone (**16**), 23-ethyl-6-O-desmethyl-auricepyrone (**18**)Aerial parts[16]***Helichrysum*** ***stoechas* (L.) Moench****Dipyrones**:


Helipyrone A (**6**)Roots[19]
Helipyrone A (**6**)Aerial parts with flowers[13,20]
Helipyrone B (Norhelipyrone) (**5**)Roots[19]
Helipyrone B (Norhelipyrone) (**5**)Aerial parts with flowers[20]
Helipyrone C (Bisnorhelipyrone) (**4**)Roots[19]
Helipyrone C (Bisnorhelipyrone) (4)Aerial parts with flowers[20]
**3-prenyl PGs**:


Arzanol (**10**)flowers[13]
**3-geranyl PGs**:


18,18-bis-desmethyl Achyroclinopyrone C (**22**)Roots[19]
18,18-bis-desmethyl Achyroclinopyrone C (**22**)Aerial parts with flowers[13]
18,18-bis-desmethyl Achyroclinopyrone A (**23**)Roots[19]
**2-prenyl PGs**:


Arenol A (**29**)Aerial parts with flowers[20]
Homoarenol (**30**)Aerial parts with flowers[15,20]
**Benzopyranes**:


Plicatipyrone (**33**), Italipyrone (**36**)Aerial parts with flowers[20]***Helichrysum*** ***stoechas* subsp. *barrelieri* (Ten.) Nym****Benzofuranes**:


20-(3,3′-Dimethylallyl)-italipyrone (**39**)Flower heads[7]***Helichrysum*** ***zeyheri* Less.**



**Monopyrones**:


3,5-dimethyl-4-hydroxy-6-isopropyl alphapyrone (**1**), 3,5-dimethyl-4-(methoxy)-6-isopropyl alphapyrone (**2**)Aerial parts[16]


## 4. Monopyrones

Monopyrones consist of a 4-hydroxy-2*H*-pyran-2-one (or α-pyrone) core, substituted with methyl groups at C3 and C5, and an alkyl side chain at C6 (Figure 2). The hydroxyl group at C4 may also be methylated.

Three monopyrone derivatives have been isolated from *Helichrysum* species collected in South Africa and Italy. Two related α-pyrones, **3,5-dimethyl-4-hydroxy-6-isopropyl alphapyrone** (**1**) (C_10_H_14_O_3_, exact mass 182.0943 Da) and its methylated analogue **3,5-dimethyl-4-(methoxy)-6-isopropyl alphapyrone** (**2**) (C_11_H_16_O_3_, exact mass 196.1099 Da), were co-isolated from the aerial parts of *Helichrysum zeyheri* collected in Transvaal, South Africa [16]. The study reported their extraction from dry aerial parts using an ether-petrol mixture at room temperature. Subsequent purification involving methanolic removal of nonpolar constituents, followed by purification steps, afforded the compounds in 0.0133% (*w*/*w*) and 0.0333% (*w*/*w*) yields, respectively. **Micropyrone** (**3**) (C_14_H_20_O_4_, exact mass 252.1362 Da) was isolated in several studies from aerial parts with flowers of *H. italicum* subspecies, collected from Sardinia, Italy. Reported yielded were 0.0054% (*w*/*w*) [10], 0.013% (*w*/*w*) [9] and 0.0482% (*w*/*w*) [8]. From aerial parts with flowers of *H. italicum* (Roth) G. Don, another study [14] obtained a 0.0015% (*w*/*w*) yield of **micropyrone** (**3**).

**Micropyrone** (**3**) has been evaluated for its anti-inflammatory [8], antioxidant [17], antibacterial [9,14], and cytotoxic [14] properties. The overall pharmacological profile suggested no meaningful bioactivity potential under the tested conditions. Taken together, the data indicate that **micropyrone** (**3**) lacks significant biological potential.

## 5. Dipyrones

Dipyrones consist of two alpha pyrones usually linked through a methylene bridge (Figure 3). Four structurally related dipyrones (**helipyrone A**, **B**, **C**, and **helitalone A)** were isolated from several *Helichrysum* species, gathered across Europe and Iran. Dipyrones consist of two substituted pyrones linked with a methylene, or other alkyl, bridge.

**Helipyrone C** (**4**) (**bisnorhelipyrone**) (C_15_H_16_O_6_, exact mass 292.0947 Da) was obtained from dried aerial parts of *H. oocephalum*, collected in Mashhad, Iran, yielding 0.00005% (*w*/*w*) [11]. The compound was also isolated from roots [19] and aerial parts with flowers [20] of *H. stoechas*, and from *H. arenarium* roots [5], though quantitative yields were not reported. **Helipyrone B** (**5**) (**norhelipyrone**) (C_16_H_18_O_6_, exact mass 306.1103 Da) was isolated from the same species and locations as **helipyrone C** (**4**). From *H. oocephalum*, the dried aerial parts yielded 0.00001% (*w*/*w*) [11]. Yields of **helipyrone B** (**5**) from *H. stoechas* and *H. arenarium* were not reported [5,19,20]. **Helipyrone A** (**6**) (C_17_H_20_O_6_, exact mass 320.126 Da) has been the most frequently isolated compound, with yields ranging from 0.00007% to 0.041% (*w*/*w*). The highest yield (0.041% *w*/*w*) was achieved from *H. stoechas* flowers using cold methanol maceration, yielding 14.5 mg from 35 g [13]. From *H. italicum* ssp. *microphyllum*, yields from aerial parts with flowers were 0.0256% (*w*/*w*) [9], 0.0245% (*w*/*w*) [8], and 0.022% (*w*/*w*) [10]. *H. oocephalum* aerial parts yielded 2.7 mg from 4 kg (0.00007% *w*/*w*) [11]. From *H. italicum*, Opitz et al., 1970 [18] reported 0.0025% (*w*/*w*). **Helitalone A** (**7**) (C_22_H_30_O_5_, 374.2093 Da) was obtained from *H. italicum* (Roth) G. Don (non-woody aerial parts with flowers, Sardinia) by acetone extraction and multi-step chromatographic purification, yielding 5 mg (0.001% *w*/*w*) from 500 g dried plant material [14].

**Helipyrone A** (**6**), **helipyrone B** (**5**), and **helipyrone C** (**4**) were evaluated for antiparasitic [22], anti-inflammatory [8], antioxidant [17], and antibacterial activities [9,12,20]. The compounds display limited therapeutic potential. **Helipyrone A** (**6**) demonstrated stronger anti-inflammatory activity than arzanol through NF-κB inhibition [8], though it was less effective at inhibiting the inflammatory enzymes mPGES-1 and 5-LOX [23]. The compound showed selective antioxidant properties, effectively preventing lipid oxidation in certain conditions but not in iron-catalyzed systems [17]. **Helipyrones A**, **B**, and **C** showed limited antibacterial activity, primarily against Gram-positive bacteria. A mixture of **Helipyrones A**, **B**, and **C** was active against *Staphylococcus aureus* and *Bacillus subtilis* but ineffective against Gram-negative bacteria [20]. Pure **helipyrone A** (**6**) displayed variable activity against drug-resistant *S. aureus* strains [9] but failed to prevent biofilm formation [12]. **Helitalone A** (**7**) showed no biological activity. The compound failed to inhibit various pathogenic bacteria including *Mycobacterium tuberculosis*, *S. aureus*, and *Enterococcus* species [14]. It also lacked cytotoxic effects against cancer cells, indicating that **Helitalone A** (**7**) was essentially inactive in both antibacterial and anticancer applications [14].

## 6. 3-NA PGs

These compounds consist of 3-non-alkylated (3-NA) acylphloroglucinol, connected with a substituted α-pyrone ring via a methylene bridge, hence classified as 3-non-alkylated phloroglucinols (3-NA PGs) (Figure 4).

In a single study, both **Norauricepyrone** (**8**) (C_20_H_24_O_7_, exact mass 376.1522 Da) and **Methyl-norauricepyrone** (**9**) (C_21_H_26_O_7_, exact mass 390.1673 Da) were co-isolated from the roots of *H. cephaloideum* DC. collected in Transvaal, South Africa [16]. Both compounds were afforded in identical, low yields of approximately 0.001% (*w*/*w*). No biological activity has been reported for 3-NA PGs.

## 7. 3-Prenyl PGs

3-prenyl PGs consist of 3-prenyl acylphloroglucinol, connected with a substituted alpha pyrone ring via a methylene bridge (Figure 5). A diverse array of C3-prenylated PG-α-pyrones, including **arzanol** (**10**), **arenol B** (**11**), **arenol C** (**14**), **heliarzanol** (**17**), **helitalone B** (**12**), and several **auricepyrone** and **norauricepyrone** derivatives (**compounds 13, 15, 16, 18**), have been reported from various *Helichrysum* species, collected across the Mediterranean and southern African localities. All 3-prenyl PGs share a similar structure, differing in the aliphatic groups in C5 of the alpha pyrone ring, or at the C1 acyl group. Only **heliarzanol** (**17**) has a hydroxyl group in the prenyl side chain at C3.

Among 3-prenyl PGs, **arzanol** (**10**) (C_22_H_26_O_7_, exact mass 402.1678 Da) is the most extensively studied, having been isolated from both *H. italicum* ssp. *microphyllum* and *H. italicum* (Roth) G. Don in Sardinia, Italy, and *H. stoechas* in Zaragoza, Spain. The highest yield (0.48% *w*/*w*) was obtained from *H. stoechas* (flowers, Zaragoza, Spain) using cold methanol maceration [13], yielding 170 mg from 35 g of plant material, while *H. italicum* ssp. *microphyllum* (Sardinia, Italy) yielded 3700 mg from purifying approximately a quarter of the residue obtained from extracting 5 kg of dried aerial parts with flowers (0.295% *w*/*w*) [9]. Other Sardinian sources of dried aerial parts with flowers from *H. italicum* ssp. *microphyllum* reported yields of 0.0965% (*w*/*w*) [8] and 0.081% (*w*/*w*) [10] for arzanol yield. The lowest yield (0.002%) was recorded from a study on dried aerial parts with flowers of *H. italicum* (Roth) G. Don [14]. **Helitalone B** (**12**) (C_23_H_28_O_7_, exact mass 416.1835 Da) was isolated from *H. italicum* in a low yield of 0.00062% (*w*/*w*) [14]. From the mother liquors of a large-scale arzanol isolation, the oxidized derivative **heliarzanol** (**17**) (C_24_H_30_O_8_, exact mass 446.1941 Da) was obtained in a yield of 0.00036% (*w*/*w*) [9]. **Arenol B** (**11**) (C_23_H_28_O_7_, exact mass 416.1835 Da) yielded 0.4 mg from 4 kg *H. oocephalum* aerial parts (0.00001% *w*/*w*) [11], while **Helitalone B** (**12**) (C_23_H_28_O_7_, exact mass 416.1835 Da) yielded 3.1 mg from 500 g *H. italicum* (0.00062% *w*/*w*) [14]. From *H. oocephalum* aerial parts collected in Mashhad, Iran, **arenol C** (**14**) and **3-prenyl norauricepyrone** (**15**) were isolated (both compounds: C_24_H_30_O_7_, exact mass 430.1991 Da), yielding 0.00002% (*w*/*w*) and 0.00001% (*w*/*w*), respectively [11]. Several other compounds were isolated from South African *Helichrysum* species. **6-O-Desmethylauricepyrone** (**13**) (C_24_H_30_O_7_, exact mass 430.1991 Da) was isolated from *H. odoratissimum* with a notable yield of 0.133% (*w*/*w*) [7]. Its homologue, **23-Methyl-6-O-desmethylauricepyrone** (**16**) (C_25_H_32_O_7_, exact mass 444.2148 Da) was reported in 0.033% (*w*/*w*) yield from *H. odoratissimum* [7], 0.07% (*w*/*w*) from *H. mixtum* roots, and 0.09% (*w*/*w*) from *H. stenopterum* aerial parts [16]. The same study [16] also reported the isolation of its analogue, **23-ethyl-6-O-desmethyl-auricepyrone** (**18**) (C_26_H_34_O_7_, exact mass 458.2304 Da), from *H. mixtum* (0.07% *w*/*w*) roots and *H. stenopterum* (0.09% *w*/*w*) aerial parts.

Regarding biological activities of 3-prenyl PGs, **arzanol** (**10**) demonstrates potent anti-inflammatory effects by inhibiting NF-κB signaling (IC_50_ ≈ 12 µM), suppressing cytokines including IL-1β, TNF-α, IL-6, and IL-8 [8]. It dually inhibits mPGES-1 (IC_50_ = 0.4 µM) and 5-LOX (IC_50_ = 3.1 µM) [23,24]. These effects were validated in vivo using rat pleurisy models, where 3.6 mg/kg significantly reduced inflammatory cell infiltration [24]. **Arzanol** (**10**) showed potent antibacterial activity against drug-resistant *S. aureus* strains in one study [9], though another study reported low antibacterial activity against the same pathogen [14]. **Arzanol** (**10**) prevents copper-induced LDL oxidation starting at 8 µM, preserving polyunsaturated fatty acids [25]. It protected VERO cells against lipid peroxidation [17] and prevented H_2_O_2_-induced apoptosis in keratinocytes and neuroblastoma cells [26,27]. In vivo studies on rat models confirmed reduction in oxidative markers at 9 mg/kg [10]. In a mouse model, feeding with *H. stoechas* extract containing **arzanol** (**10**) (100 mg/kg for 3 weeks) produced anxiolytic effects comparable to diazepam and antidepressant effects comparable to amitriptyline, without impairing locomotor activity or memory [28]. The extract attenuated weight gain of mice without reducing food intake, suggesting metabolic modulation [28]. **Arzanol** (**10**) protected against glutamate-induced excitotoxicity at 5–10 μM, indicating neuroprotective properties [28]. **Arzanol** (**10**) exhibited selective cytotoxicity against cancer cells, particularly Caco-2 colon cancer cells, while showing no toxicity below 100 μM in normal cells, suggesting a favorable therapeutic window [10].

Several other 3-prenyl PG compounds showed promising antiparasitic and antiviral potential. **3-prenyl norauricepyrone** (**15**) was identified as a potential HIV-Reverse Transcriptase inhibitor through virtual screening, demonstrating strong binding affinity to key active site residues [29]. **Arenol C** (**14**) and **arenol B** (**11**) exhibited antiprotozoal activity with IC_50_ values of 4.8–5.8 µM against *Leishmania* and 5.5–8.0 µM against *Plasmodium falciparum*, showing good selectivity, though considerably weaker than clinical standards miltefosine and chloroquine. **23-Methyl-6-O-desmethylauricepyrone** (**16**) also demonstrated notable antimalarial activity (IC_50_ 2.5 µM) but with lower selectivity, while **3-prenyl norauricepyrone** (**15**) showed moderate antiprotozoal effects. **Helitalone B** (**12**) was biologically inactive, displaying no antibacterial or cytotoxic properties at tested concentrations.

## 8. 3-Prenyl Methoxy PGs

In contrast to 3-prenyl PGs, 3-prenyl methoxy PGs have an esterified phloroglucinol ring (Figure 6). Three 3-prenyl methoxy phloroglucinol derivatives were isolated from South African *Helichrysum* species.

**Auricepyrone** (**19**) (C_25_H_32_O_7_, exact mass 444.2148 Da) was first isolated from the roots of *H. auriceps*, where extraction with a diethyl ether/petroleum ether mixture afforded a yield of 0.015% (*w*/*w*) [6]. The same study [6] reported a slightly lower yield of 0.0125% (*w*/*w*) from the roots of *H. cephaloideum* [6]. The aerial parts contained a significantly lower concentration of 0.0025% (*w*/*w*) [6]. Later studies on *H. cephaloideum* roots confirmed the presence of **auricepyrone** (**19**), with one reporting a yield of 0.0079% (*w*/*w*) [16], while another study confirmed its presence qualitatively without reporting a yield [7]. **23-Methylauricepyrone** (**20**) (C_26_H_34_O_7_, exact mass 458.2304 Da) was found in 0.045% (*w*/*w*) yield from the roots of *H. auriceps* and 0.0375% (*w*/*w*) from the roots of *H. cephaloideum* [6]. The aerial parts of *H. cephaloideum* contained only 0.0075% (*w*/*w*) [6]. The presence of **23-Methylauricepyrone** (**20**) in *H. cephaloideum* roots was also confirmed qualitatively, without specifying yield, in another report [7]. The series extends further with the isolation of **23-ethyl-6-O-desmethyl-4-O-methylauricepyrone** (**21**) (C_27_H_36_O_7_, exact mass 472.2461 Da). This analogue was co-isolated with **auricepyrone** (**19**) from the roots of *H. cephaloideum*, exhibiting a yield of 0.0079% (*w*/*w*) [16]. These results indicate that 3-prenyl methoxy phloroglucinols (PGs) were present in higher quantities in *H. auriceps* than in *H. cephaloideum*, with greater accumulation observed in the roots compared to the aerial parts, under comparable extraction conditions. No biological activity is reported for 3-prenyl methoxy PGs.

## 9. 3-Geranyl PGs

This subclass possesses a geranyl, rather than prenyl, side chain at C3 of the acylphenone moiety (Figure 7). Seven 3-geranyl-substituted phloroglucinol derivatives have been isolated from *Helichrysum* species native to southern Europe and Iran. Since the study that described the isolation of compounds **22–24** [21] did not assign them a trivial name, in this study, compounds were assigned the trivial names **18,18-bis-desmethyl achyroclinopyrone C** (**22**)**, 18,18-bis-desmethyl achyroclinopyrone A** (**23**), and **8′-methyl-18,18-bis-desmethyl achyroclinopyrone A** (**24**), based on their structural similarity to the closest related analogues with established trivial names.

**18,18-bis-desmethyl achyroclinopyrone C** (**22**) (C_26_H_32_O_7_, exact mass 456.2148 Da) was isolated at 0.0857% (*w*/*w*) yield from the flowers and aerial parts of *H. stoechas* from Spain, achieved through cold methanol maceration [13]. In contrast, a lower yield of 0.0032% (*w*/*w*) was obtained from the aerial parts of *H. decumbens*, also from Spain, using a brief room-temperature chloroform extraction [21]. The presence of this compound was also confirmed in the roots of *H. stoechas* from France, although its yield was not reported [19]. Its analogue, **18,18-bis-desmethyl achyroclinopyrone A** (**23**) (C_27_H_34_O_7_, exact mass 470.2304 Da) was quantified in *H. decumbens* at a 0.0008% (*w*/*w*) yield [21] and qualitatively identified in the roots of *H. stoechas* [19]. **8′-methyl-18,18-bis-desmethyl Achyroclinopyrone A** (**24**) (C_28_H_36_O_7_, exact mass 484.2461 Da) was detected only in trace amounts in *H. decumbens* [21]. Other C3 geranyl analogues, **achyroclinopyrones A–D**, have been discovered much later in a single study in 2019 [11], exclusively reported from the aerial parts of *H. oocephalum*, collected from Iran. A single, large-scale methanol percolation, followed by extensive liquid–liquid partitioning and multi-step chromatographic purification, led to the isolation of the four **achyroclinopyrones**, all in low yields [11]. The isolated metabolites include **achyroclinopyrone C** (**25**) (C_28_H_36_O_7_, exact mass 484.2461 Da) at 0.00003% (*w*/*w*), **achyroclinopyrone A** (**26**) (C_29_H_38_O_7_, exact mass 498.2617 Da) at 0.00004% (*w*/*w*), its isobar **achyroclinopyrone D** (**27**) at 0.00005% (*w*/*w*), and **achyroclinopyrone B** (**28**) (C_30_H_40_O_7_, exact mass 512.2774 Da) at 0.00003% (*w*/*w*).

**18,18-bis-desmethyl achyroclinopyrone C** (**22**) showed moderate antibacterial activity against Gram-positive bacteria (*Bacillus* sp. and *Staphylococcus epidermidis*) but was inactive against Gram-negative bacteria [30]. It selectively inhibited the fungus *Cladosporium herbarum* with weaker effects on other fungi [21]. Virtual screening also identified this compound as a potential HIV-Reverse Transcriptase inhibitor [29].

The **achyroclinopyrones A–D** demonstrated antiprotozoal activity [11]. **Achyroclinopyrone B** (**28**) was most active against malaria parasites (*P. falciparum*), while **achyroclinopyrone A** (**26**) showed strongest activity against *Leishmania*. Although less potent than clinical drugs chloroquine and miltefosine, these compounds displayed favorable selectivity with low toxicity to mammalian cells.

## 10. 2-Prenyl PGs

This subclass, in contrast to all other presented PGs have different structural arrangements for the prenyl side chain, the acyl group, and the α-pyrone ring (Figure 8).

**Arenol A** (**29**) (C_21_H_24_O_7_, exact mass 388.1522 Da) was obtained from aerial parts with flowers of *H. stoechas*, collected in Spain [20]. The same compound was also reported from *H. arenarium* (non-woody aerial parts with flowers) in an earlier study [5]. Similarly, **Homoarenol** (**30**) (C_22_H_26_O_7_) was isolated from *H. stoechas* [5,20]. These 2-prenyl PGs may represent rare positional isomers within the *Helichrysum* metabolome, potentially relevant for structural diversification and bioactivity profiling. **Arenol A** (**29**) and its homologue **homoarenol** (**30**), have been qualitatively described, though quantitative data is not reported [20].

The **arenol/homoarenol mixture** showed weak antibacterial activity limited to Gram-positive bacteria [20]. It inhibited *Bacillus subtilis*, *S. aureus*, and *S. epidermidis* (MIC 12 mg/L) with weaker activity against *M. phlei* (MIC 25 mg/L). The mixture was completely inactive against Gram-negative bacteria including *E. coli*, *Klebsiella*, *Pseudomonas*, and *Salmonella*. These MIC values are substantially higher than clinically useful antibiotics, limiting therapeutic potential.

## 11. Amino PGs

A structurally unique class of phloroglucinol derivatives, the **helichrytalicines**, has been isolated from Italian *H. italicum* [12]. These compounds are biosynthetically distinct from the more common phloroglucinol-α-pyrone conjugates described so far, as they possess amino groups at the para-position of the acylphloroglucinol ring and within the C1 acyl side chain, representing a distinct structural type for this genus (Figure 9).

In a single study [12], **helichrytalicine A** (**31**) (C_20_H_26_N_2_O_6_, exact mass 390.1791 Da) and its lower homologue **helichrytalicine B** (**32**) (C_19_H_24_N_2_O_6_, exact mass 376.1634 Da) were co-isolated from the leaf material of *H. italicum* collected in Castel Volturno, Italy [12]. The isolation procedure involved room-temperature ultrasound-assisted extraction with methanol. The resulting crude extract was subjected to liquid–liquid partitioning, and the ethyl acetate fraction was then purified through a multi-step process involving size-exclusion chromatography, silica-gel column chromatography and preparative TLC. This process yielded 6 mg of **helichrytalicine A** (**31**) and 2.3 mg of **Helichrytalicine B** (**32**) from the same extract, although the initial starting mass of the plant material was not reported. The discovery of these amino-phloroglucinols significantly broadens the known metabolic diversity of *H. italicum*.

**Helichrytalicine A** (**31**) and **helichrytalicine B** (**32**) showed selective antibacterial activity against Gram-positive bacteria [12]. **Helichrytalicine B** (**32**) exhibited potent bactericidal activity against *S. epidermidis*, reducing planktonic growth by 77% and inhibiting 80% of biofilm formation at 128 µg/mL. In contrast, **helichrytalicine A** (**31**) showed only moderate bacteriostatic activity (70% growth reduction) with no significant anti-biofilm effects. Neither compound was active against the Gram-negative bacterium *P. aeruginosa*, confirming their selectivity for Gram-positive infections similar to other phloroglucinol compounds.

## 12. Benzopyranes

A distinct group of phloroglucinol-derived metabolites, characterized by the presence of a benzopyrane structural motif, has been reported from several geographically diverse *Helichrysum* species (Figure 10).

**Plicatipyrone** (**33**) (C_22_H_26_O_8_, exact mass 418.1628 Da) was identified in two separate studies, from Mediterranean and Anatolian *Helichrysum* species. Its quantitative isolation was documented from the flower heads of *H. plicatum* collected in Turkey. A large-scale extraction of 4.5 kg of plant material, involving sequential extraction with petroleum ether and chloroform followed by silica-gel chromatography, yielded 10 mg of the compound, corresponding to a yield of 0.00022% (*w*/*w*) [7]. **Plicatipyrone** (**33**) was also identified in the aerial parts and flowers of *H. stoechas* from Spain. In this case, the plant material was macerated with dichloromethane, and the compound was isolated after chromatographic purification; however, the yield was not reported [20]. From the South African species *H. mixtum*, two related benzopyranes were isolated, showing differential accumulation between the plant’s roots and aerial parts. From the roots, **isobutyryl-helichromenopyrone** (**34**) (C_24_H_30_O_7_, exact mass 428.1835 Da) was obtained with a yield of 0.01% (*w*/*w*). In contrast, its homologue **methylbutyryl-helichromenopyrone** (**35**) (C_25_H_30_O_7_, exact mass 442.1991 Da) was isolated from the aerial parts with a lower yield of 0.001% (*w*/*w*). Both compounds were extracted using an ether-petrol mixture and purified via a combination of methanolic cleanup and silica gel chromatography [16].

**Plicatipyrone** (**33**) demonstrated moderate antibacterial activity selective for Gram-positive bacteria [20]. It inhibited *B. subtilis* and *M. phlei* at 6 mg/L, with weaker activity against *S. aureus* and *S. epidermidis* (MIC 12 mg/L). The compound was completely inactive against Gram-negative bacteria including *E. coli*, *Klebsiella*, *Pseudomonas*, and *Salmonella* at concentrations up to 100 mg/L, confirming its narrow spectrum of activity.

## 13. Benzofuranes

A diverse group of phloroglucinol derivatives featuring a benzofurane moiety (Figure 11) has been isolated from a wide geographical range of *Helichrysum* species, spanning Europe, Turkey, South Africa, and Iran.

**Italipyrone** (**36**) (C_22_H_24_O_7_, exact mass 400.1522 Da) was isolated from commercially sourced *H. italicum* involving methanol maceration and Soxhlet extraction (yield of 0.000068% *w*/*w*) [7]. Its presence was also confirmed qualitatively in the aerial parts of *H. stoechas* from Spain, although quantitative data was not provided [20]. A prenylated derivative, **20-(3,3′-Dimethylallyl)-italipyrone** (**38**) (C_27_H_32_O_7_, exact mass 468.2148 Da), was isolated from the flower heads of *H. stoechas* subsp. *barrelieri* from Turkey. Following petroleum ether extraction and extensive chromatographic purification, 80 mg was obtained from 1.6 kg of dried flower heads in a 0.005% (*w*/*w*) yield [7]. A distinct chemotaxonomic profile is evident in South African *Helichrysum* species, which produce a homologous series of alkylated **italipyrones**, primarily found in the root tissues. **22-Methyl-22-ethyl-italipyrone** (**37**) (C_25_H_30_O_7_, exact mass 442.1991 Da) was isolated from the roots of *H. cephaloideum* in yields of 0.005% (*w*/*w*) [7] and 0.0079% (*w*/*w*) [16]. Notably, its accumulation was ten-fold higher in the roots of *H. mixtum*, which afforded a 0.05% (*w*/*w*) yield [16]. Similarly, **22-Methyl-22-propyl-italipyrone** (**39**) (C_26_H_32_O_7_, exact mass 456.2148 Da) was found in *H. cephaloideum* roots (0.005% *w*/*w*) [7] and aerial parts (0.0075% *w*/*w*) [16] but was again most concentrated in the roots of *H. mixtum*, providing a 0.05% (*w*/*w*) yield [16].

Further illustrating the genus’ biosynthetic diversity, a unique cyclized benzofurane, **helicyclol** (**40**) (C_28_H_34_O_7_, exact mass 482.2304 Da), was isolated from the aerial parts of *H. oocephalum* from Iran. It was obtained after a large-scale methanol percolation and extensive purification of 4 kg of initial dried aerial parts, but only 0.5 mg was isolated (0.000012% *w*/*w*), marking it as a minor but structurally significant metabolite [11].

**Italipyrone** (**36**) showed strong antibacterial activity against Gram-positive bacteria with MICs of 3 mg/L for *B. subtilis* and *M. phlei*, and 6 mg/L for *Staphylococcus* species, but was inactive against Gram-negative bacteria [20]. Virtual screening identified **italipyrone** (**36**) as a promising HIV-Reverse Transcriptase inhibitor through strong binding to active site residues [29]. **Helicyclol** (**40**) demonstrated antiprotozoal activity with IC_50_ values of 2.0 µM against *Leishmania donovani* and 7.12 µM against *Plasmodium falciparum* [22]. While showing moderate selectivity against *Leishmania* (SI = 6.0), its activity was 7-fold weaker than miltefosine and substantially less potent than chloroquine for malaria. The narrow selectivity index against *Plasmodium* (SI = 1.7) limits its antimalarial therapeutic potential.

## 14. Chromane PGs

A pair of α-pyrone derivatives featuring a chromane ring system (Figure 12), **cycloarzanol C** (**41**) and **helicepyrone** (**42**), have been co-isolated from the Iranian species *H. oocephalum*. The discovery of these compounds highlights another distinct structural class of metabolites produced by this particular species, known for its diverse phytochemical profile.

Both **Cycloarzanol C** (**41**) (C_21_H_22_O_7_, exact mass 386.1365 Da) and **helicepyrone** (**42**) (C_29_H_36_O_7_, exact mass 496.2461 Da) were obtained from the aerial parts of the plant in a single study [11]. **Cycloarzanol C** (**41**) was isolated at 0.000013% (*w*/*w*), and **helicepyrone** (**42**) at 0.000025% (*w*/*w*), both from the dichloromethane-soluble fraction of the dried MeOH extract.

**Cycloarzanol C** (**41**) and **Helicepyrone** (**42**) showed comparable antiprotozoal activity but differed significantly in their therapeutic potential [11]. Both compounds demonstrated potent antileishmanial activity with nearly identical IC_50_ values (1.8–1.9 µM) against *L. donovani*, though 6-fold weaker than miltefosine. Their antimalarial activity was more modest (IC_50_ 9.3–11.3 µM against *P. falciparum*) and substantially weaker than chloroquine. The critical distinction was their selectivity profiles. **Cycloarzanol C** (**41**) exhibited low cytotoxicity (IC_50_ 53.1 µM), yielding an excellent selectivity index of 29.7 for antileishmanial activity. In contrast, **helicepyrone** (**42**) was twice as cytotoxic (IC_50_ 21.4 µM), resulting in a lower selectivity index of 11.2. Both compounds showed poor selectivity for antimalarial effects. Despite equal antiparasitic potency, **cycloarzanol C** (**41**) showed significantly better safety profile and selectivity.

## 15. Hetero-Trimer PGs

A pair of structurally complex hetero-trimeric phloroglucinol derivatives (Figure 13), **italidipyrone** (**43**) (C_28_H_32_O_10_, exact mass 528.1995 Da) and **23-methyl-italidipyrone** (**44**) (C_29_H_34_O_10_, 542.2152 Da) were isolated from two geographically and taxonomically distinct *Helichrysum* species.

**Italidipyrone** (**43**) was first reported from *H. italicum* (aerial parts and flowers, commercial source, Italy) [7] isolated in a 0.0001% (*w*/*w*) yield. **Italidipyrone** (**43**) was later also isolated from *H. oocephalum* (Iran), yielding 0.00002% (*w*/*w*) [11]. **23-Methyl-italidipyrone** (**44**) was likewise reported from both species. In *H. italicum*, it was obtained at 0.00002% (*w*/*w*) under identical extraction conditions as **italidipyrone** (**43**) [7]. From *H. oocephalum*, the same compound was isolated at 1.5 mg (0.00004% *w*/*w*) from 4 kg of aerial parts using methanol percolation and similar chromatographic purification steps [11].

**Italidipyrone** (**43**) demonstrated potent and selective antileishmanial activity with an IC_50_ of 2.5 µM against *L. donovani* [11]. Though 9-fold weaker than miltefosine, its low mammalian cell cytotoxicity (IC_50_ 57.6 µM) offered an excellent selectivity index of 22.9. Its antimalarial activity was more modest (IC_50_ 8.6 µM) with poor selectivity (SI 4.4). Virtual screening identified **italidipyrone** (**43**) and **23-Methyl-italicidipyrone** (**44**) as top candidates for HIV-Reverse Transcriptase inhibition [29]. **Italidipyrone** (**43**) ranked first with a docking score of −12.153 kcal/mol, followed by **23-Methyl-italicidipyrone** (**44**) at −11.977 kcal/mol.

## 16. Spiroketals

**Helispiroketals A–H** constitute a structurally distinct subclass of phloroglucinol derivatives featuring 3*H*,3′*H*-spiro[benzofuran-2,2′-furan]-3′-one scaffold (Figure 14). These compounds were isolated from *H. oocephalum* Boiss. (Mashhad, Iran) following large-scale methanol percolation (4 kg, 20 L, 24 h, RT), solvent partitioning, and RP-HPLC purification [11].

**Helispiroketal A** (**45**) (C_20_H_22_O_6_, exact mass 358.1416 Da) was isolated at 1.9 mg (0.000048% *w*/*w*), while **helispiroketal B** (**46**) and **helispiroketal F** (**47**) (C_21_H_24_O_6_, 372.1573 Da) yielded 0.9 mg (0.000023% *w*/*w*) and 1.2 mg (0.00003% *w*/*w*), respectively. **Helispiroketal E** (**48**) (C_22_H_26_O_6_, exact mass 386.1729 Da) and **helispiroketal D** (**50**) (C_23_H_28_O_6_, exact mass 400.1886 Da) were each isolated at 1 mg (0.000025% *w*/*w*), and **helispiroketal C** (**49**) (C_23_H_28_O_6_, exact mass 400.1886 Da) yielded 0.7 mg (0.000018% *w*/*w*). **Helispiroketal H** (**51**) (C_25_H_30_O_6_, exact mass 426.2042 Da) was obtained at 1.1 mg (0.000028% *w*/*w*), and **helispiroketal G** (**52**) (C_27_H_34_O_6_, exact mass 454.2355 Da) was the highest-yielding member, isolated at 2.2 mg (0.000055% *w*/*w*).

The series of eight **helispiroketal analogues (A–H)** showed antiprotozoal activity, particularly against *L. donovani* [11]. **Helispiroketals D** (**50**) and **E** (**48**) were most potent with IC_50_ values of 1.7–1.9 µM, though 6-fold weaker than miltefosine. **Helispiroketal H** (**51**) emerged as the most promising candidate, combining good antileishmanial potency (IC_50_ 2.8 µM) with the highest selectivity index (SI 15.6). **Helispiroketal A** (**45**) also showed favorable selectivity (SI 11.1) despite moderate potency (IC_50_ 5.1 µM). The series showed poor antimalarial activity, being substantially weaker than chloroquine with minimal selectivity.

## 17. Conclusions

Over the past five decades, the isolation of phloroglucinol α–pyrone derivatives from *Helichrysum* species distributed across the Mediterranean, Africa, and Asia has revealed a substantial structural and biosynthetic diversity. The 52 distinct phloroglucinol–pyrone metabolites presented in this review highlight the breadth of structural types as well as their variable accumulation patterns across organs and species. Advances in chromatographic and spectroscopic techniques—particularly NMR and high-resolution mass spectrometry—have substantially expanded the known chemical space of these metabolites, facilitating the characterization of previously elusive trace constituents. Arzanol, the most studied compound, has shown intriguing mPGES-1/5-LOX inhibition, brain glycogen phosphorylase activation, and SIRT1 inhibition. These specific activities provide defined targets for evaluating structurally related compounds, particularly those with intact C3 prenyl chains which showed enhanced anti-inflammatory and antioxidant activities in the reported assays. Priority research areas include ADME characterization and targeted screening for arzanol’s demonstrated mechanisms. Most compounds occur at yields below 0.05%, necessitating synthetic or improved biotechnological production for further development. This systematic synthesis of literature enhances the chemical understanding of *Helichrysum* phloroglucinol α–pyrones and provides a comprehensive phytochemical reference. Beyond their potential pharmacological applications, these metabolites hold significance for chemotaxonomy, bioprospecting, and standardization of *Helichrysum*-based natural products. Future research should focus on biosynthetic pathway elucidation, sustainable isolation methods, and advanced biological screening to further investigate this structurally diverse class of natural compounds.

## Figures and Tables

**Figure 1 plants-14-03460-f001:**
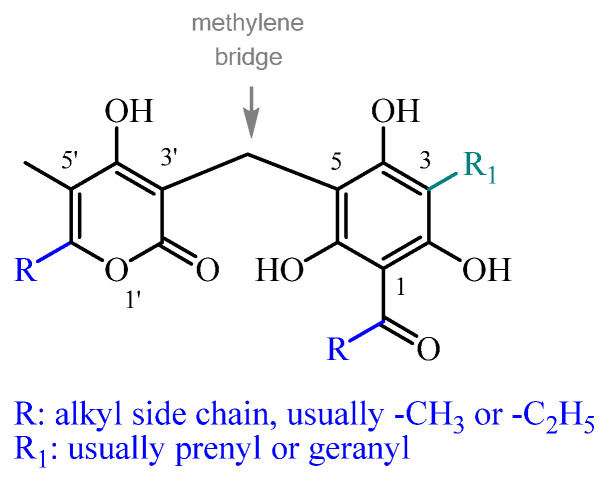
The main structural variability of α-pyrone-PGs stems from the differently substituted alklyl side chains at R.

**Figure 2 plants-14-03460-f002:**
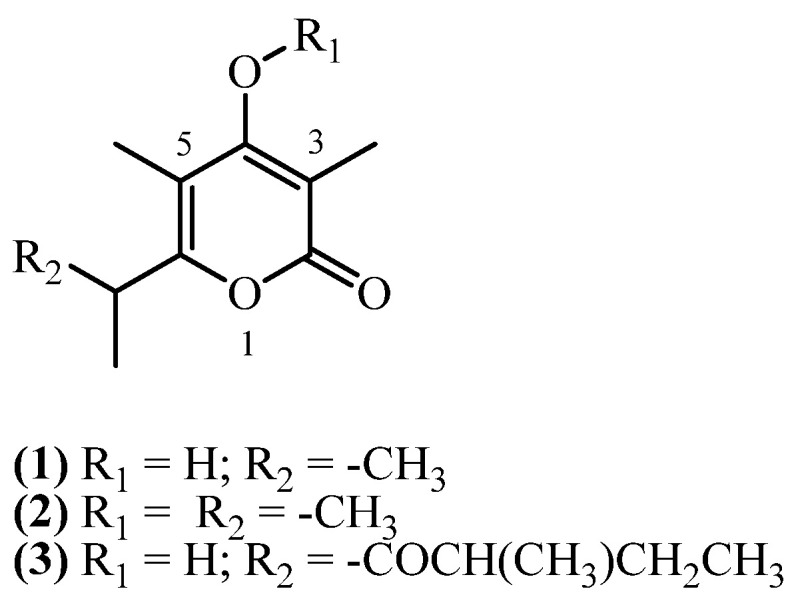
Monopyrones.

**Figure 3 plants-14-03460-f003:**
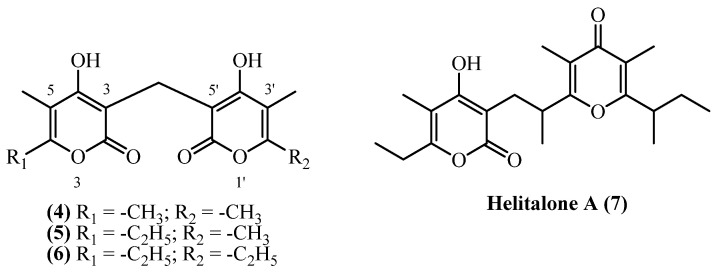
Dipyrones.

**Figure 4 plants-14-03460-f004:**
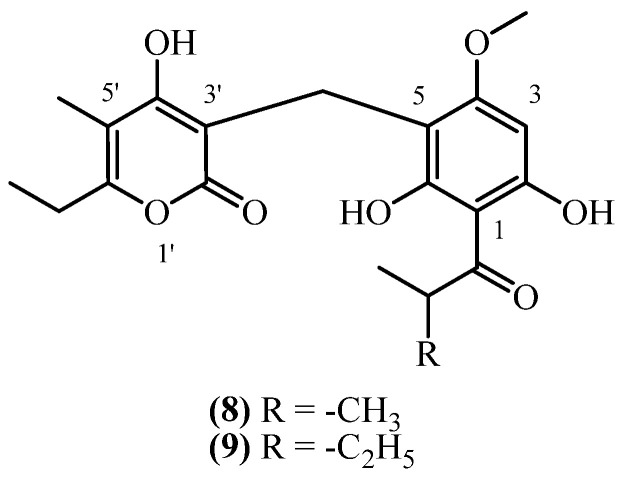
3-non-alkylated acylphloroglucinol (3-NA PGs).

**Figure 5 plants-14-03460-f005:**
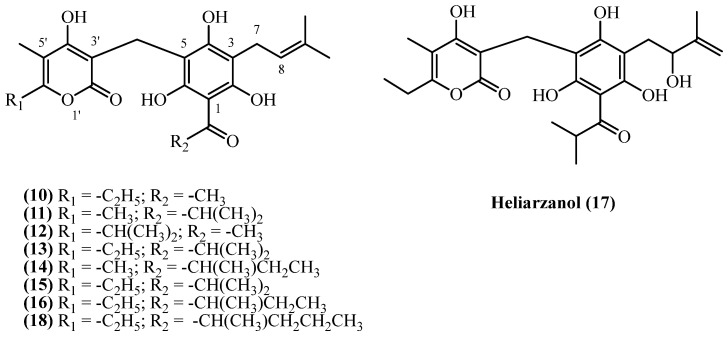
3-prenyl PGs.

**Figure 6 plants-14-03460-f006:**
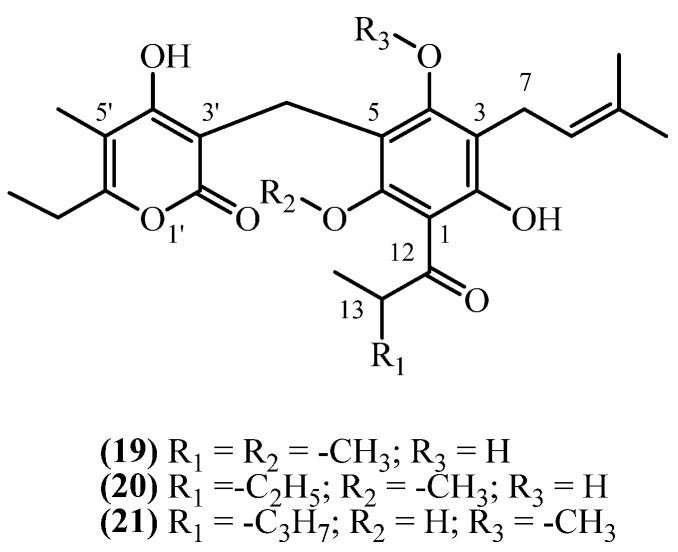
3-prenyl methoxy PGs.

**Figure 7 plants-14-03460-f007:**
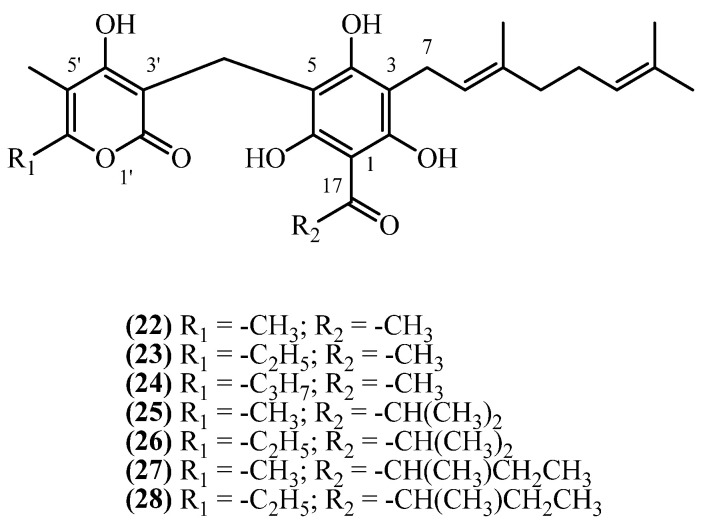
3-geranyl PGs.

**Figure 8 plants-14-03460-f008:**
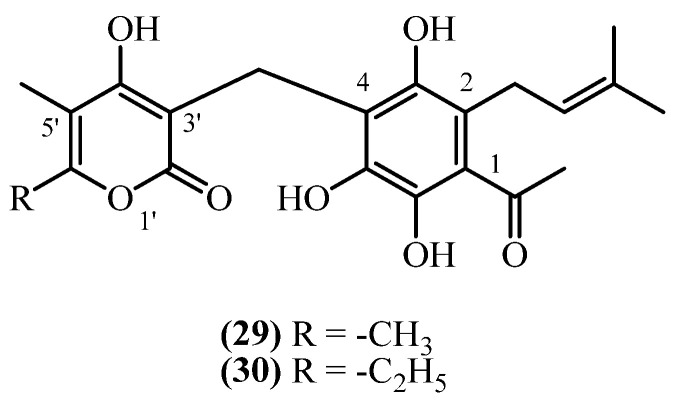
2-prenyl PGs.

**Figure 9 plants-14-03460-f009:**
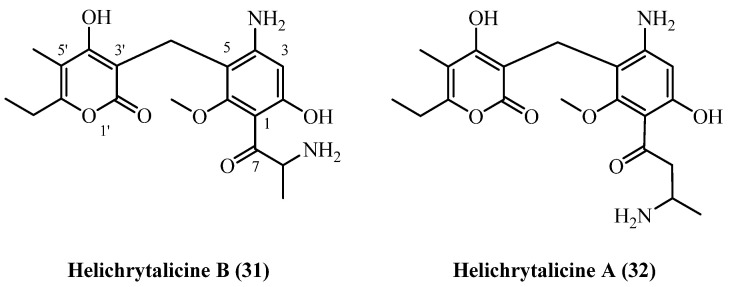
Amino PGs.

**Figure 10 plants-14-03460-f010:**
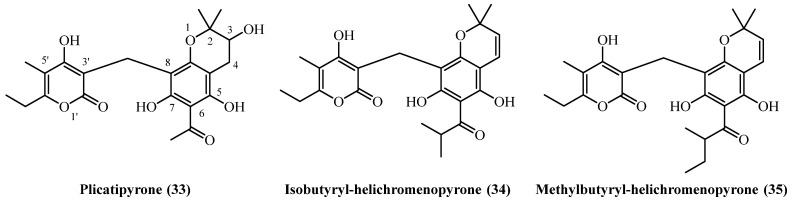
Benzopyranes.

**Figure 11 plants-14-03460-f011:**
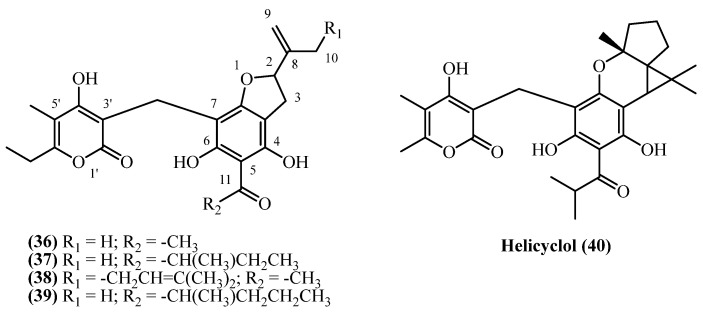
Benzofuranes.

**Figure 12 plants-14-03460-f012:**
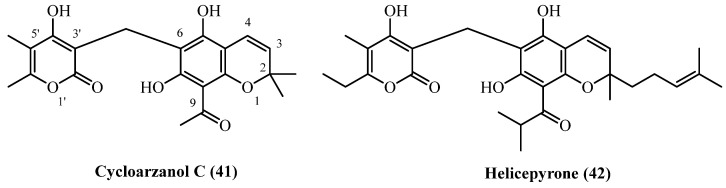
Chromane PGs.

**Figure 13 plants-14-03460-f013:**
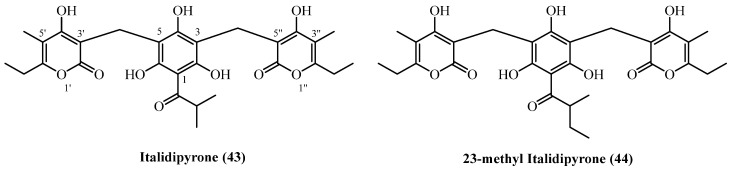
Hetero-trimer PGs.

**Figure 14 plants-14-03460-f014:**
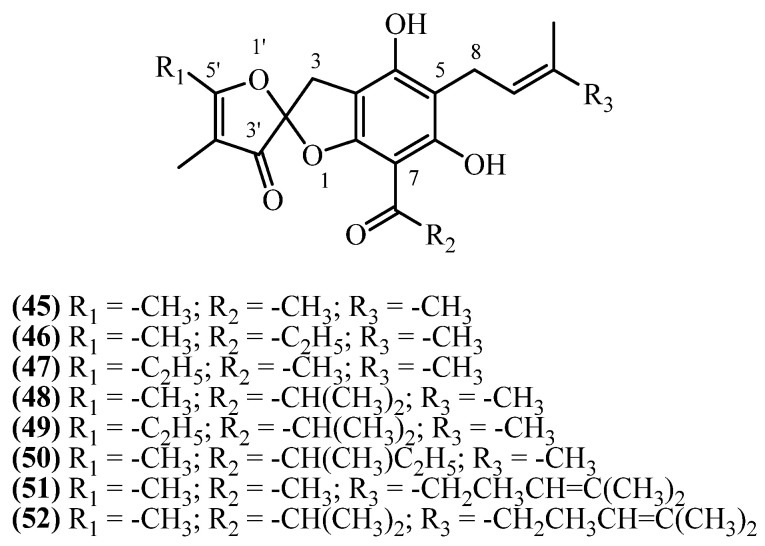
Spiroketals.

**Table 2 plants-14-03460-t002:** Extraction conditions, yields, and structural classification of α-pyrone compounds isolated from *Helichrysum* species.

Entry	Compound Name	Extraction Conditions	Extracted Amount	Plant Name, Location	Plant Parts	Subclass	Ref.
**1**	3,5-dimethyl-4-hydroxy-6-isopropyl alphapyrone (**1**)	Extracted with Et2O-petrol (1:2) at rt. The extracts obtained after removal of long chain saturated carbons with MeOH, were first separated by CC (silica gel) and further by TLC (silica gel PF 254)	Plant material: 150 g Amount isolated: 20 mg Yield: 0.0133%	Plant name: *Helichrysum zeyheri* Less. Location: Transvaal, South Africa (wildgrown)	Aerial parts	Monopyrones	[16]
**2**	3,5-dimethyl-4-(methoxy)-6-isopropyl alphapyrone (**2**)	The same as for entry **1**, compound **1**	Plant material: 150 g Amount isolated: 50 mg Yield: 0.0333%	Plant name: *Helichrysum zeyheri* Less. Location: Transvaal, South Africa (wildgrown)	Aerial parts	Monopyrones	[16]
**3**	Micropyrone (**3**)	Maceration with acetone (3 × 4 L) at rt; residue (51 g) adsorbed on silica gel and fractionated by vacuum chromatography into petroleum-ether, EtOAc, and acetone fractions; the EtOAc eluate crystallized from EtOAc/petroleum ether	Plant material: 1000 g Amount isolated: 482 mg Yield: 0.0482%	Plant name: *Helichrysum italicum* ssp. *Microphyllum* Location: near Arzana (NU, Sardinia), Italy	Aerial parts (non-woody) + flowers	Monopyrones	[8]
**4**	Micropyrone (**3**)	Maceration with acetone (3 × 2 L, rt); crude residue (12 g) dissolved in EtOAc/hexane (1:1, 200 mL each) and left overnight; precipitate collected and recrystallised from toluene–acetone. Later silica-gel fractions (hexane/EtOAc 6:4) triturated with diethyl ether	Plant material: 540 g Amount isolated: 29 mg Yield: 0.0054%	Plant name: *Helichrysum italicum* ssp. *Microphyllum* Location: Sardinia, Italy (wildgrown)	Aerial parts + flowers	Monopyrones	[17]
**5**	Micropyrone (**3**)	Acetone extraction (2 × 15 L, rt); 50 g of the crude residue (from a total of 202 g gummy residue) was partitioned between petroleum ether and 90% MeOH (1 h, 30 °C); fractions of the aq MeOH phase chromatographed on silica gel (petroleum ether→EtOAc); finally—trituration with ether	Plant material: 5000 g Amount isolated: 162.5 mg Yield: 0.0129%	Plant name: *Helichrysum italicum* ssp. *Microphyllum* Location: Arzana (Sardinia), Italy (wildgrown)	Aerial parts + flowers	Monopyrones	[9]
**6**	Micropyrone (**3**)	Acetone extraction (3 × 5 L, rt); crude residue (20 g) fractionated by silica-gel vacuum liquid chromatography (hexane → EtOAc → CH_2_Cl_2_ → MeOH); chromatographed on Sephadex LH-20 and semi-prep HPLC	Plant material: 500 g Amount isolated: 7.5 mg Yield: 0.0015%	Plant name: *Helichrysum italicum* (Roth) G.Don Location: Sardinia, Italy (wildgrown)	non-woody Aerial parts + flowers	Monopyrones	[14]
**7**	Helipyrone C (Bisnorhelipyrone) (**4**)	Extracted with petroleum ether and ethanol, followed by solvent partitioning; CC on silica gel using chloroform–methanol (98:2), and after acetylation, the compounds were further purified by silica gel chromatography (hexane–ether, 9:1) and preparative TLC to yield diacetyl derivatives, which in turn, after deacetylation, provided helipyrone (**6**), bisnorhelipyrone (**4**), and norhelipyrone (**5**)	Plant material: 1500 g Amount isolated: n.d. Yield: n.d.	Plant name: *Helichrysum arenarium* (L.) Moench Location: near Malacky, Slovakia (wildgrown)	Roots	Dipyrones	[5]
**8**	Helipyrone C (Bisnorhelipyrone) (**4**)	Methanol percolation (20 L, 24 h, rt); crude MeOH extract (55 g dry extract) partitioned with petroleum ether, CH_2_Cl_2_, EtOAc, n-BuOH; CH_2_Cl_2_ fraction (12 g) silica-gel CC (n-hexane → CHCl_3_) then RP-HPLC (MeCN/TFA 55→100%)	Plant material: 4000 g Amount isolated: 1.9 mg Yield: 4.8 × 10^−5^%	Plant name: *Helichrysum oocephalum* Boiss. Location: Mashhad, Razavi Khorasan, Iran	Aerial parts	Dipyrones	[11]
**9**	Helipyrone C (Bisnorhelipyrone) (**4**)	Successive extraction with hexane, CHCl3, EtOAc, MeOH at rt; silica-gel column chromatography (hexane–EtOAc); preparative TLC (CHCl_3_–MeOH 99:1)	Plant material: 400 g Amount isolated: n.d. Yield: n.d.	Plant name: *Helichrysum stoechas* (L.) Moench Location: Dunes of the forest domain of Olonne, near Les Sables d’Olonne, Vendée, France	Roots	Dipyrones	[19]
**10**	Helipyrone C (Bisnorhelipyrone) (**4**)	Maceration of dried aerial parts with CH_2_Cl_2_ (3 × to exhaustion, RT); CH_2_Cl_2_ extract concentrated in vacuo, chromatographed on silica-gel 60 (CHCl_3_/MeOH gradient) then silica-gel 60 H; active fractions purified by preparative TLC	Plant material: 700 g Amount isolated: n.d. Yield: n.d.	Plant name: *Helichrysum stoechas* (L.) Moench Location: Sierra Perenchiza, near Chiva, Valencia, Spain	Aerial parts + flowers	Dipyrones	[20]
**11**	Helipyrone B (Norhelipyrone) (**5**)	Same as for entry **7**, compound **4**	Plant material: 1500 g Amount isolated: n.d. Yield: n.d.	Plant name: *Helichrysum arenarium* (L.) Moench Location: near Malacky, Slovakia (wildgrown)	Roots	Dipyrones	[5]
**12**	Helipyrone B (Norhelipyrone) (**5**)	Procedure the same as entry **8**, compound **4**	Plant material: 4000 g Amount isolated: 0.4 mg Yield: 1 × 10^−5^%	Plant name: *Helichrysum oocephalum* Boiss. Location: Mashhad, Razavi Khorasan, Iran	Aerial parts	Dipyrones	[11]
**13**	Helipyrone B (Norhelipyrone) (**5**)	Procedure the same as entry **9**, compound **4**	Plant material: 400 g Amount isolated: n.d. Yield: n.d.	Plant name: *Helichrysum stoechas* (L.) Moench Location: Dunes of the forest domain of Olonne, near Les Sables d’Olonne, Vendée, France	Roots	Dipyrones	[19]
**14**	Helipyrone B (Norhelipyrone) (**5**)	Procedure the same as entry **10**, compound **4**	Plant material: 700 g Amount isolated: n.d. Yield: n.d.	Plant name: *Helichrysum stoechas* (L.) Moench Location: Sierra Perenchiza, near Chiva, Valencia, Spain	Aerial parts + flowers	Dipyrones	[20]
**15**	Helipyrone A (**6**)	Procedure the same as entry **11**, compound **5**	Plant material: 1500 g Amount isolated: n.d. Yield: n.d.	Plant name: *Helichrysum arenarium* (L.) Moench Location: near Malacky, Slovakia (wildgrown)	Roots	Dipyrones	[5]
**16**	Helipyrone A (**6**)	Procedure the same as entry **3**, compound **3**	Plant material: 1000 g Amount isolated: 245 mg Yield: 0.0245%	Plant name: *Helichrysum italicum* ssp. *microphyllum* Location: near Arzana (NU, Sardinia), Italy	Aerial parts (non-woody) + flowers	Dipyrones	[8]
**17**	Helipyrone A (**6**)	Maceration with acetone (3 × 2 L, RT); crude residue (12 g) dissolved in EtOAc/hexane (1:1, 200 mL each) and left overnight; precipitate collected and recrystallised from toluene–acetone. Fractionation on silica-gel column (hexane → EtOAc)	Plant material: 540 g Amount isolated: 120 mg Yield: 0.0219%	Plant name: *Helichrysum italicum* ssp. *microphyllum* Location: Sardinia, Italy (wildgrown)	Aerial parts + flowers	Dipyrones	[17]
**18**	Helipyrone A (**6**)	Same as entry **5**, compound **3**; Aqueous MeOH phase fraction crystallized from diethyl ether to give helipyrone	Plant material: 5000 g Amount isolated: 320 mg Yield: 0.0256%	Plant name: *Helichrysum italicum* ssp. *microphyllum* Location: Arzana (Sardinia), Italy (wildgrown)	Aerial parts + flowers	Dipyrones	[9]
**19**	Helipyrone A (**6**)	Methanol ultrasound extraction of dried leaves (3 × 1 h, RT); crude residue (38.7 g) partitioned between H_2_O and EtOAc; EtOAc layer (8.8 g) fractionated on Sephadex LH-20 (hexane/CHCl_3_/MeOH 3:1:1; EtOAc fractions from methanol extract subjected to silica-gel CC (2-butanone/hexane 1:4) and preparative TLC	Plant material: n.d. Amount isolated: 1.2 mg Yield: n.d.	Plant name: *Helichrysum italicum* (Roth) G.Don Location: Castel Volturno (Caserta, Italy) (wildgrown)	Leaf material	Dipyrones	[12]
**20**	Helipyrone A (**6**)	Methanol maceration; Celite adsorption; Soxhlet extraction with petroleum ether; The residue was purified by silica-gel chromatography (CH_2_Cl_2_–MeOH gradient)	Plant material: 37,000 g Amount isolated: n.d. Yield: n.d.	Plant name: *Helichrysum italicum* (Roth) G.Don Location: Italy (commerial source)	Aerial parts + flowers	Dipyrones	[7]
**21**	Helipyrone A (**6**)	Defatted by refluxing in petroleum ether (2 h); extracted three times with 80:20 ethanol–water (3 × 5 L, RT); combined hydroalcoholic extracts partitioned between H_2_O and CHCl_3_; CHCl_3_ phase concentrated to dryness; residue purified by silica-gel column chromatography (CHCl_3_→MeOH gradient) and preparative TLC	Plant material: 10,000 g Amount isolated: 250 mg Yield: 0.0025%	Plant name: *Helichrysum italicum* (Roth) G.Don Location: Italy (commerial source)	Aerial parts	Dipyrones	[18]
**22**	Helipyrone A (**6**)	Procedure the same as entry **8**, compound **4**	Plant material: 4000 g Amount isolated: 2.7 mg Yield: 6.79 × 10^−5^%	Plant name: *Helichrysum oocephalum* Boiss. Location: Mashhad, Razavi Khorasan, Iran	Aerial parts	Dipyrones	[11]
**23**	Helipyrone A (**6**)	Procedure the same as entry **9**, compound **4**	Plant material: 400 g Amount isolated: n.d. Yield: n.d.	Plant name: *Helichrysum stoechas* (L.) Moench Location: Dunes of the forest domain of Olonne, near Les Sables d’Olonne, Vendée, France	Roots	Dipyrones	[19]
**24**	Helipyrone A (**6**)	Cold maceration (3 × 24 h, 4 °C) with MeOH (1 L each); combined extracts evaporated to dryness (3.5 g); after CHCl_3_/MeOH (95:5→60:40) silica-gel CC, purified on Sephadex LH-20	Plant material: 35 g Amount isolated: 14.5 mg Yield: 0.041%	Plant name: *Helichrysum stoechas* (L.) Moench Location: Villanueva de Gállego (Zaragoza, Spain)	Aerial parts + flowers	Dipyrones	[13]
**25**	Helipyrone A (**6**)	Procedure the same as entry **10**, compound **4**	Plant material: 700 g Amount isolated: n.d. Yield: n.d.	Plant name: *Helichrysum stoechas* (L.) Moench Location: Sierra Perenchiza, near Chiva, Valencia, Spain	Aerial parts + flowers	Dipyrones	[20]
**26**	Helitalone A (**7**)	Procedure the same as entry **6**, compound **3**	Plant material: 500 g Amount isolated: 5 mg Yield: 0.001%	Plant name: *Helichrysum italicum* (Roth) G.Don Location: Sardinia, Italy (wildgrown)	Non-woody Aerial parts + flowers	Dipyrones	[14]
**27**	Norauricepyrone (**8**)	Procedure the same as entry **1**, compound **1**	Plant material: 190 g Amount isolated: 2 mg Yield: 0.001052%	Plant name: *Helichrysum cephaloideum* DC. Location: Transvaal, South Africa (wildgrown)	Roots	3-NA PGs	[16]
**28**	Methyl-norauricepyrone (**9**)	Procedure the same as entry **1**, compound **1**	Plant material: 190 g Amount isolated: 2 mg Yield: 0.001052%	Plant name: *Helichrysum cephaloideum* DC. Location: Transvaal, South Africa (wildgrown)	Roots	3-NA PGs	[16]
**29**	Arzanol (**10**)	Procedure the same as entry **16**, compound **6**	Plant material: 1000 g Amount isolated: 965 mg Yield: 0.0965%	Plant name: *Helichrysum italicum* ssp. *microphyllum* Location: near Arzana (NU, Sardinia), Italy	Aerial parts (non-woody) + flowers	3-prenyl PGs	[8]
**30**	Arzanol (**10**)	Maceration with acetone (3 × 2 L, RT); crude residue (12 g) dissolved in EtOAc/hexane (1:1, 200 mL each) and left overnight; precipitate collected and recrystallised from toluene–acetone; mother liquors chromatographed on silica gel (hexane → EtOAc gradient)	Plant material: 540 g Amount isolated: 437 mg Yield: 0.081%	Plant name: *Helichrysum italicum* ssp. *microphyllum* Location: Sardinia, Italy (wildgrown)	Aerial parts + flowers	3-prenyl PGs	[17]
**31**	Arzanol (**10**)	Same as entry **5**, compound **3**; aqueous MeOH phase fraction crystallized from ether, then silica-gel CC (petroleum ether/EtOAc)	Plant material: 5000 g Amount isolated: 3700 mg Yield: 0.296%	Plant name: *Helichrysum italicum* ssp. *microphyllum* Location: Arzana (Sardinia), Italy (wildgrown)	Aerial parts + flowers	3-prenyl PGs	[9]
**32**	Arzanol (**10**)	Procedure the same as entry **6**, compound **3**	Plant material: 500 g Amount isolated: 10 mg Yield: 0.002%	Plant name: *Helichrysum italicum* (Roth) G. Don Location: Sardinia, Italy (wildgrown)	Non-woody Aerial parts + flowers	3-prenyl PGs	[14]
**33**	Arzanol (**10**)	Cold maceration (3 × 24 h, 4 °C) with MeOH (1 L each); combined extracts evaporated to dryness (3.5 g) and chromatographed on silica gel (n-BuOH:H_2_O 82:18 → 70:10:30) then Sephadex LH-20	Plant material: 35 g Amount isolated: 170 mg Yield: 0.48%	Plant name: *Helichrysum stoechas* (L.) Moench Location: Villanueva de Gállego (Zaragoza, Spain)	Flowers	3-prenyl PGs	[13]
**34**	Arenol B (**11**)	Procedure the same as entry **8**, compound **4**	Plant material: 4000 g Amount isolated: 0.4 mg Yield: 1 × 10^−5^%	Plant name: *Helichrysum oocephalum* Boiss. Location: Mashhad, Razavi Khorasan, Iran	Aerial parts	3-prenyl PGs	[11]
**35**	Helitalone B (**12**)	Procedure the same as entry **6**, compound **3**	Plant material: 500 g Amount isolated: 3.1 mg Yield: 0.00062%	Plant name: *Helichrysum italicum* (Roth) G. Don Location: Sardinia, Italy (wildgrown)	Non-woody Aerial parts + flowers	3-prenyl PGs	[14]
**36**	6-O-Desmethylauricepyrone (**13**)	Ether: petroleum-ether (3:1, RT) extraction of air-dried aerial parts; silica-gel column (Activity Stage II) followed by repeated preparative TLC (Si-gel GF254)	Plant material: 90 g Amount isolated: 120 mg Yield: 0.133%	Plant name: *Helichrysum odoratissimum* Sweet. Location: South Africa (wildgrown)	Aerial parts	3-prenyl PGs	[7]
**37**	Arenol C (**14**)	Procedure the same as entry **8**, compound **4**	Plant material: 4000 g Amount isolated: 0.8 mg Yield: 2 × 10^−5^%	Plant name: *Helichrysum oocephalum* Boiss. Location: Mashhad, Razavi Khorasan, Iran	Aerial parts	3-prenyl PGs	[11]
**38**	3-prenyl norauricepyrone (**15**)	Procedure the same as entry **8**, compound **4**	Plant material: 4000 g Amount isolated: 0.5 mg Yield: 1.25 × 10^−5^%	Plant name: *Helichrysum oocephalum* Boiss. Location: Mashhad, Razavi Khorasan, Iran	Aerial parts	3-prenyl PGs	[11]
**39**	23-Methyl-6-O-desmethylauricepyrone (**16**)	Procedure the same as entry **1**, compound **1**	Plant material: 10 g Amount isolated: 7 mg Yield: 0.0699%	Plant name: Helichrysum mixtum (Kuntze) Moeser Location: Transvaal, South Africa (wildgrown)	Roots	3-prenyl PGs	[16]
**40**	23-Methyl-6-O-desmethylauricepyrone (**16**)	Procedure the same as entry **36**, compound **13**	Plant material: 90 g Amount isolated: 30 mg Yield: 0.033%	Plant name: *Helichrysum odoratissimum* Sweet. Location: South Africa (wildgrown)	Aerial parts	3-prenyl PGs	[7]
**41**	23-Methyl-6-O-desmethylauricepyrone (**16**)	Procedure the same as entry **8**, compound **4**	Plant material: 4000 g Amount isolated: 0.3 mg Yield: 7.5 × 10^−6^%	Plant name: *Helichrysum oocephalum* Boiss. Location: Mashhad, Razavi Khorasan, Iran	Aerial parts	3-prenyl PGs	[11]
**42**	23-Methyl-6-O-desmethylauricepyrone (**16**)	Procedure the same as entry **1**, compound **1**	Plant material: 100 g Amount isolated: 90 mg Yield: 0.09%	Plant name: *Helichrysum stenopterum* DC. Location: Transvaal, South Africa (wildgrown)	Aerial parts	3-prenyl PGs	[16]
**43**	Heliarzanol (**17**)	Same as entry **31**, compound **10**; Mother liquors from arzanol isolation chromatographed on silica gel (hexane/EtOAc 55:45) and preparative HPLC	Plant material: 5000 g Amount isolated: 4.5 mg Yield: 0.00036%	Plant name: *Helichrysum italicum* ssp. *microphyllum* Location: Arzana (Sardinia), Italy (wildgrown)	Aerial parts + flowers	3-prenyl PGs	[9]
**44**	23-ethyl-6-O-desmethyl-auricepyrone (**18**)	Procedure the same as entry **1**, compound **1**	Plant material: 10 g Amount isolated: 7 mg Yield: 0.0699%	Plant name: *Helichrysum mixtum* (Kuntze) Moeser Location: Transvaal, South Africa (wildgrown)	Roots	3-prenyl PGs	[16]
**45**	23-ethyl-6-O-desmethyl-auricepyrone (**18**)	Procedure the same as entry **1**, compound **1**	Plant material: 100 g Amount isolated: 90 mg Yield: 0.09%	Plant name: *Helichrysum stenopterum* DC. Location: Transvaal, South Africa (wildgrown)	Aerial parts	3-prenyl PGs	[16]
**46**	Auricepyrone (**19**)	Extracted with diethyl ether/petroleum ether (1:2) at rt; combined extracts separated on silica gel column chromatography (activity grade II), further purified by repeated TLC (Si gel GF254)	Plant material: 50 g Amount isolated: 7.5 mg Yield: 0.01499%	Plant name: *Helichrysum auriceps* Hilliard Location: Natal (South Africa)	Roots	3-prenyl methoxy PGs	[6]
**47**	Auricepyrone (**19**)	Same as entry **46**, compound **19**	Plant material: 30 g Amount isolated: 3.75 mg Yield: 0.0125%	Plant name: *Helichrysum cephaloideum* DC. Location: Natal (South Africa)	Roots	3-prenyl methoxy PGs	[6]
**48**	Auricepyrone (**19**)	Same as entry **46**, compound **19**	Plant material: 50 g Amount isolated: 1.25 mg Yield: 0.0025%	Plant name: *Helichrysum cephaloideum* DC. Location: Natal (South Africa)	Aerial parts	3-prenyl methoxy PGs	[6]
**49**	Auricepyrone (**19**)	Ether: petroleum-ether (1:7, RT) extraction of shredded roots; silica-gel column (Activity Stage II) followed by repeated preparative TLC (Si-gel GF254)	Plant material: 50 g Amount isolated: n.d. Yield: n.d.	Plant name: *Helichrysum cephaloideum* DC. Location: South Africa (wildgrown)	Roots	3-prenyl methoxy PGs	[7]
**50**	Auricepyrone (**19**)	Procedure the same as entry **1**, compound **1**	Plant material: 190 g Amount isolated: 15 mg Yield: 0.007894%	Plant name: *Helichrysum cephaloideum* DC. Location: Transvaal, South Africa (wildgrown)	Roots	3-prenyl methoxy PGs	[16]
**51**	23-Methylauricepyrone (**20**)	Procedure the same as entry **46**, compound **19**	Plant material: 50 g Amount isolated: 22.5 mg Yield: 0.0449%	Plant name: *Helichrysum auriceps* Hilliard Location: Natal (South Africa)	Roots	3-prenyl methoxy PGs	[6]
**52**	23-Methylauricepyrone (**20**)	Procedure the same as entry **46**, compound **19**	Plant material: 30 g Amount isolated: 11.25 mg Yield: 0.03749%	Plant name: *Helichrysum cephaloideum* DC. Location: Natal (South Africa)	Roots	3-prenyl methoxy PGs	[6]
**53**	23-Methylauricepyrone (**20**)	Procedure the same as entry **48**, compound **19**	Plant material: 50 g Amount isolated: 3.75 mg Yield: 0.00749%	Plant name: *Helichrysum cephaloideum* DC. Location: Natal (South Africa)	Aerial parts	3-prenyl methoxy PGs	[6]
**54**	23-Methylauricepyrone (**20**)	Procedure the same as entry **49**, compound **19**	Plant material: 50 g Amount isolated: n.d. Yield: n.d.	Plant name: *Helichrysum cephaloideum* DC. Location: South Africa (wildgrown)	Roots	3-prenyl methoxy PGs	[7]
**55**	23-ethyl-6-O-desmethyl-4-O-methylauricepyrone (**21**)	Procedure the same as entry **1**, compound **1**	Plant material: 190 g Amount isolated: 15 mg Yield: 0.007894%	Plant name: *Helichrysum cephaloideum* DC. Location: Transvaal, South Africa (wildgrown)	Roots	3-prenyl methoxy PGs	[16]
**56**	18,18-bis-desmethyl Achyroclinopyrone C (**22**)	Chloroform extraction (plant dipped for 10 min at RT); silica-gel CC (n-hexane–EtOAc); preparative TLC (CHCl_3_); semipreparative reversed-phase HPLC (MeOH–phosphate buffer)	Plant material: 500 g Amount isolated: 16 mg Yield: 0.0032%	Plant name: *Helichrysum decumbens* Cambess. Location: La Manga del Mar Menor, Murcia, Spain (wildgrown)	Aerial parts + flowers	3-geranyl PGs	[21]
**57**	18,18-bis-desmethyl Achyroclinopyrone C (**22**)	Successive extraction with hexane, CHCl_3_, EtOAc, MeOH at rt; silica-gel CC (hexane–EtOAc); preparative TLC (CHCl_3_–MeOH 99:1)	Plant material: 400 g Amount isolated: n.d. Yield: n.d.	Plant name: *Helichrysum stoechas* (L.) Moench Location: Dunes of the forest domain of Olonne, near Les Sables d’Olonne, Vendée, France	Roots	3-geranyl PGs	[19]
**58**	18,18-bis-desmethyl Achyroclinopyrone C (**22**)	Procedure the same as entry **24**, compound **6** Obtained from silica-gel fractions after polarity-gradient CH_2_Cl_2_/MeOH chromatography, separated from arzanol by recrystallisation	Plant material: 35 g Amount isolated: 30 mg Yield: 0.08569%	Plant name: *Helichrysum stoechas* (L.) Moench Location: Villanueva de Gállego (Zaragoza, Spain)	Aerial parts + flowers	3-geranyl PGs	[13]
**59**	18,18-bis-desmethyl Achyroclinopyrone A (**23**)	Procedure the same as entry **56**, compound **22**	Plant material: 500 g Amount isolated: 4 mg Yield: 0.0008%	Plant name: *Helichrysum decumbens* Cambess. Location: La Manga del Mar Menor, Murcia, Spain (wildgrown)	Aerial parts + flowers	3-geranyl PGs	[21]
**60**	18,18-bis-desmethyl Achyroclinopyrone A (**23**)	Procedure the same as entry **9**, compound **4**	Plant material: 400 g Amount isolated: n.d. Yield: n.d.	Plant name: *Helichrysum stoechas* (L.) Moench Location: Dunes of the forest domain of Olonne, near Les Sables d’Olonne, Vendée, France	Roots	3-geranyl PGs	[19]
**61**	8′-methyl-18,18-bis-desmethyl Achyroclinopyrone A (**24**)	Procedure the same as entry **56**, compound **22** Detected as trace amounts by reversed-phase HPLC (MeOH–phosphate buffer)	Plant material: 500 g Amount isolated: traces Yield: n.d.	Plant name: *Helichrysum decumbens* Cambess. Location: La Manga del Mar Menor, Murcia, Spain (wildgrown)	Aerial parts + flowers	3-geranyl PGs	[21]
**62**	Achyroclinopyrone C (**25**)	Same as entry **8**, compound **4**	Plant material: 4000 g Amount isolated: 1.1 mg Yield: 2.8 × 10^−5^%	Plant name: *Helichrysum oocephalum* Boiss. Location: Mashhad, Razavi Khorasan, Iran	Aerial parts	3-geranyl PGs	[11]
**63**	Achyroclinopyrone A (**26**)	Procedure the same as entry **62**, compound **25**	Plant material: 4000 g Amount isolated: 1.5 mg Yield: 3.75 × 10^−5^%	Plant name: *Helichrysum oocephalum* Boiss. Location: Mashhad, Razavi Khorasan, Iran	Aerial parts	3-geranyl PGs	[11]
**64**	Achyroclinopyrone D (**27**)	Procedure the same as entry **62**, compound **25**	Plant material: 4000 g Amount isolated: 2.1 mg Yield: 5.3 × 10^−5^%	Plant name: *Helichrysum oocephalum* Boiss. Location: Mashhad, Razavi Khorasan, Iran	Aerial parts	3-geranyl PGs	[11]
**65**	Achyroclinopyrone B (**28**)	Procedure the same as entry **62**, compound **25**	Plant material: 4000 g Amount isolated: 1.2 mg Yield: 2.99 × 10^−5^%	Plant name: *Helichrysum oocephalum* Boiss. Location: Mashhad, Razavi Khorasan, Iran	Aerial parts	3-geranyl PGs	[11]
**66**	Arenol A (**29**)	n.d.	Plant material: n.d. Amount isolated: n.d. Yield: n.d.	Plant name: *Helichrysum arenarium* (L.) Moench Location: n.d.	Non-woody Aerial parts + flowers	2-prenyl PGs	[15]
**67**	Arenol A (**29**)	Procedure the same as entry **10**, compound **4**	Plant material: 700 g Amount isolated: n.d. Yield: n.d.	Plant name: *Helichrysum stoechas* (L.) Moench Location: Sierra Perenchiza, near Chiva, Valencia, Spain	Aerial parts + flowers	2-prenyl PGs	[20]
**68**	Homoarenol (**30**)	Procedure the same as entry **10**, compound **4**	Plant material: 700 g Amount isolated: n.d. Yield: n.d.	Plant name: *Helichrysum stoechas* (L.) Moench Location: Sierra Perenchiza, near Chiva, Valencia, Spain	Aerial parts + flowers	2-prenyl PGs	[20]
**69**	Homoarenol (**30**)	n.d.	Plant material: n.d. Amount isolated: n.d. Yield: n.d.	Plant name: *Helichrysum stoechas* (L.) Moench Location: n.d.	non-woody Aerial parts + flowers	2-prenyl PGs	[15]
**70**	Helichrytalicine B (**31**)	Methanol ultrasound extraction of dried leaves (3 × 1 h, RT); crude residue (38.7 g) partitioned between H_2_O and EtOAc; EtOAc layer (8.8 g) fractionated on Sephadex LH-20 (hexane/CHCl3/MeOH 3:1:1), followed by silica-gel CC and preparative TLC	Plant material: n.d. Amount isolated: 2.3 mg Yield: n.d.	Plant name: *Helichrysum italicum* (Roth) G.Don Location: Castel Volturno (Caserta, Italy) (wildgrown)	Leaf material	Amino PGs	[12]
**71**	Helichrytalicine A (**32**)	Procedure the same as entry **70**, compound **31**	Plant material: n.d. Amount isolated: 6 mg Yield: n.d.	Plant name: *Helichrysum italicum* (Roth) G.Don Location: Castel Volturno (Caserta, Italy) (wildgrown)	Leaf material	Amino PGs	[12]
**72**	Plicatipyrone (**33**)	Extracted with petroleum ether, followed by EtOH partition and chloroform extraction. The residue was purified by silica-gel CC using cyclohexane:EtOAc (8:2)	Plant material: 4500 g Amount isolated: 10 mg Yield: 0.000222%	Plant name: *Helichrysum plicatum* DC. ssp. *plicatum* Location: Yildiz Dagi and Camlibel (Sivas), Turkey (wildgrown)	Flower heads	Benzopyranes	[7]
**73**	Plicatipyrone (**33**)	Procedure the same as entry **10**, compound **4**	Plant material: 700 g Amount isolated: n.d. Yield: n.d.	Plant name: *Helichrysum stoechas* (L.) Moench Location: Sierra Perenchiza, near Chiva, Valencia, Spain	Aerial parts + flowers	Benzopyranes	[20]
**74**	Isobutyryl-helichromenopyrone (**34**)	Procedure the same as entry **1**, compound **1**	Plant material: 10 g Amount isolated: 1 mg Yield: 0.01%	Plant name: *Helichrysum mixtum* (Kuntze) Moeser Location: Transvaal, South Africa (wildgrown)	Roots	Benzopyranes	[16]
**75**	methylbutyryl-helichromenopyrone (**35**)	Procedure the same as entry **1**, compound **1**	Plant material: 100 g Amount isolated: 1 mg Yield: 0.001%	Plant name: *Helichrysum mixtum* (Kuntze) Moeser Location: Transvaal, South Africa (wildgrown)	Aerial parts	Benzopyranes	[16]
**76**	Italipyrone (**36**)	Procedure the same as entry **20**, compound **6**	Plant material: 37,000 g Amount isolated: 25 mg Yield: 6.799 × 10^−5^%	Plant name: *Helichrysum italicum* (Roth) G.Don Location: Italy (commerial source)	Aerial parts + flowers	Benzofuranes	[7]
**77**	Italipyrone (**36**)	Procedure the same as entry **10**, compound **4**	Plant material: 700 g Amount isolated: n.d. Yield: n.d.	Plant name: *Helichrysum stoechas* (L.) Moench Location: Sierra Perenchiza, near Chiva, Valencia, Spain	Aerial parts + flowers	Benzofuranes	[20]
**78**	22-Methyl-22-ethyl-italipyrone (**37**)	Same as entry **49**, compound **19**	Plant material: 50 g Amount isolated: 2.5 mg Yield: 0.005%	Plant name: *Helichrysum cephaloideum* DC. Location: South Africa (wildgrown)	Roots	Benzofuranes	[7]
**79**	22-Methyl-22-ethyl-italipyrone (**37**)	Procedure the same as entry **1**, compound **1**	Plant material: 190 g Amount isolated: 15 mg Yield: 0.007894%	Plant name: *Helichrysum cephaloideum* DC. Location: Transvaal, South Africa (wildgrown)	Roots	Benzofuranes	[16]
**80**	22-Methyl-22-ethyl-italipyrone (**37**)	Procedure the same as entry **1**, compound **1**	Plant material: 10 g Amount isolated: 5 mg Yield: 0.05%	Plant name: *Helichrysum mixtum* (Kuntze) Moeser Location: Transvaal, South Africa (wildgrown)	Roots	Benzofuranes	[16]
**81**	22-Methyl-22-propyl-italipyrone (**38**)	Same as entry **49**, compound **19**; Isolated after de-acetylation	Plant material: 50 g Amount isolated: 2.5 mg Yield: 0.005%	Plant name: *Helichrysum cephaloideum* DC. Location: South Africa (wildgrown)	Roots	Benzofuranes	[7]
**82**	22-Methyl-22-propyl-italipyrone (**38**)	Procedure the same as entry **1**, compound **1**	Plant material: 200 g Amount isolated: 15 mg Yield: 0.007499%	Plant name: *Helichrysum cephaloideum* DC. Location: Transvaal, South Africa (wildgrown)	Aerial parts	Benzofuranes	[16]
**83**	22-Methyl-22-propyl-italipyrone (**38**)	Procedure the same as entry **1**, compound **1**	Plant material: 10 g Amount isolated: 5 mg Yield: 0.05%	Plant name: *Helichrysum mixtum* (Kuntze) Moeser Location: Transvaal, South Africa (wildgrown)	Roots	Benzofuranes	[16]
**84**	20-(3,3′-Dimethylallyl)-italipyrone (**39**)	Extracted with petroleum ether; partitioned with EtOH; the dry residue extracted with chloroform. The extract (8.9 g) was purified by silica-gel CC, followed by rechromatography using cyclohexane:ethyl acetate (9:1)	Plant material: 1600 g Amount isolated: 80 mg Yield: 0.005%	Plant name: *Helichrysum stoechas* subsp. *barrelieri* (Ten.) Nym Location: Mudanya (Bursa), Turkey (wildgrown)	Flower heads	Benzofuranes	[7]
**85**	Helicyclol (**40**)	Procedure the same as entry **62**, compound **25**	Plant material: 4000 g Amount isolated: 0.5 mg Yield: 1.2 × 10^−5^%	Plant name: *Helichrysum oocephalum* Boiss. Location: Mashhad, Razavi Khorasan, Iran	Aerial parts	Benzofuranes	[11]
**86**	Cycloarzanol C (**41**)	Procedure the same as entry **62**, compound **25**	Plant material: 4000 g Amount isolated: 0.5 mg Yield: 1.299 × 10^−5^%	Plant name: *Helichrysum oocephalum* Boiss. Location: Mashhad, Razavi Khorasan, Iran	Aerial parts	Chromane PGs	[11]
**87**	Helicepyrone (**42**)	Procedure the same as entry **62**, compound **25**	Plant material: 4000 g Amount isolated: 1 mg Yield: 2.5 × 10^−5^%	Plant name: *Helichrysum oocephalum* Boiss. Location: Mashhad, Razavi Khorasan, Iran	Aerial parts	Chromane PGs	[11]
**88**	Italidipyrone (**43**)	Procedure the same as entry **20**, compound **6**	Plant material: 37,000 g Amount isolated: 36 mg Yield: 9.729 × 10^−5^%	Plant name: *Helichrysum italicum* (Roth) G.Don Location: Italy (commerial source)	Aerial parts + flowers	Hetero-trimer PGs	[7]
**89**	Italidipyrone (**43**)	Procedure the same as entry **8**, compound **4**	Plant material: 4000 g Amount isolated: 0.8 mg Yield: 2 × 10^−5^%	Plant name: *Helichrysum oocephalum* Boiss. Location: Mashhad, Razavi Khorasan, Iran	Aerial parts	Hetero-trimer PGs	[11]
**90**	23-Methyl-italidipyrone (**44**)	Procedure the same as entry **20**, compound **6**	Plant material: 37,000 g Amount isolated: 9 mg Yield: 2.43 × 10^−5^%	Plant name: *Helichrysum italicum* (Roth) G.Don Location: Italy (commerial source)	Aerial parts + flowers	Hetero-trimer PGs	[7]
**91**	23-Methyl-italidipyrone (**44**)	Procedure the same as entry **8**, compound **4**	Plant material: 4000 g Amount isolated: 1.5 mg Yield: 3.8 × 10^−5^%	Plant name: *Helichrysum oocephalum* Boiss. Location: Mashhad, Razavi Khorasan, Iran	Aerial parts	Hetero-trimer PGs	[11]
**92**	Helispiroketal A (**45**)	Procedure the same as entry **8**, compound **4**	Plant material: 4000 g Amount isolated: 1.9 mg Yield: 4.8 × 10^−5^%	Plant name: *Helichrysum oocephalum* Boiss. Location: Mashhad, Razavi Khorasan, Iran	Aerial parts	Spiroketals	[11]
**93**	Helispiroketal B (**46**)	Procedure the same as entry **8**, compound **4**	Plant material: 4000 g Amount isolated: 0.9 mg Yield: 2.3 × 10^−5^%	Plant name: *Helichrysum oocephalum* Boiss. Location: Mashhad, Razavi Khorasan, Iran	Aerial parts	Spiroketals	[11]
**94**	Helispiroketal F (**47**)	Procedure the same as entry **8**, compound **4**	Plant material: 4000 g Amount isolated: 1.2 mg Yield: 3 × 10^−5^%	Plant name: *Helichrysum oocephalum* Boiss. Location: Mashhad, Razavi Khorasan, Iran	Aerial parts	Spiroketals	[11]
**95**	Helispiroketal E (**48**)	Procedure the same as entry **8**, compound **4**	Plant material: 4000 g Amount isolated: 1 mg Yield: 2.5 × 10^−5^%	Plant name: *Helichrysum oocephalum* Boiss. Location: Mashhad, Razavi Khorasan, Iran	Aerial parts	Spiroketals	[11]
**96**	Helispiroketal C (**49**)	Procedure the same as entry **8**, compound **4**	Plant material: 4000 g Amount isolated: 0.7 mg Yield: 1.8 × 10^−5^%	Plant name: *Helichrysum oocephalum* Boiss. Location: Mashhad, Razavi Khorasan, Iran	Aerial parts	Spiroketals	[11]
**97**	Helispiroketal D (**50**)	Procedure the same as entry **8**, compound **4**	Plant material: 4000 g Amount isolated: 1 mg Yield: 2.5 × 10^−5^%	Plant name: *Helichrysum oocephalum* Boiss. Location: Mashhad, Razavi Khorasan, Iran	Aerial parts	Spiroketals	[11]
**98**	Helispiroketal H (**51**)	Procedure the same as entry **8**, compound **4**	Plant material: 4000 g Amount isolated: 1.1 mg Yield: 2.8 × 10^−5^%	Plant name: *Helichrysum oocephalum* Boiss. Location: Mashhad, Razavi Khorasan, Iran	Aerial parts	Spiroketals	[11]
**99**	Helispiroketal G (**52**)	Procedure the same as entry **8**, compound **4**	Plant material: 4000 g Amount isolated: 2.2 mg Yield: 5.5 × 10^−5^%	Plant name: *Helichrysum oocephalum* Boiss. Location: Mashhad, Razavi Khorasan, Iran	Aerial parts	Spiroketals	[11]

## Data Availability

No new data were created or analyzed in this study.

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
