# Peer review of "Phloroglucinol α-Pyrones from Helichrysum: A Review on Structural Diversity, Plant Distribution and Isolation"

_plants, 2025, doi:10.3390/plants14223460_

Round 1
Reviewer 1 Report
Comments and Suggestions for Authors
Dear Author,
I am impressed by the huge effort made by You to write the review. However I think that several items can be improved:
- In Literature Search Methodology: please write the sentence stating in what period of years the reference literature was collected.
- Please also unify the way of writing in the manuscript the names in Latin (for example all of them can be written in italics, both botanical family and plant species.
- In Suppl. Materials: why the literature covers only the period from 1970 to 2019 ? Perhaps some new references after 2019 can be added ?
Author Response
Dear Reviewer, I would like to express gratitude for your positive evaluation for this review manuscript which would indeed be an invaluable reference for future phytochemical investigation into this interesting class of phloroglucinol α-pyrones derived from Helichrysum. All made changes to the manuscript are highlighted with yellow.
Reviewer Comment 1: "In Literature Search Methodology: please write the sentence stating in what period of years the reference literature was collected."
Answer Comment 1: The first report of isolation and characterization of phloroglucinol α-pyrones date to Vrkcoč et al, 1970, and the latest - to Akaberi et al, 2019. Hence, in the "Literature Search Methodology" we have added this information.
Reviewer Comment 2: "Please also unify the way of writing in the manuscript the names in Latin (for example all of them can be written in italics, both botanical family and plant species."
Answer Comment 2: Botanical names have been unified.
Reviewer Comment 3: "In Suppl. Materials: why the literature covers only the period from 1970 to 2019 ? Perhaps some new references after 2019 can be added ?"
Answer Comment 3: All available literature irrrespective of years was searched and the latest phloroglucinols α-pyrones freom Helichrysum have been described by Akaberi et al, 2019 and Werner et al, 2019.
Reviewer 2 Report
Comments and Suggestions for Authors
The publication is very well written and clearly described. It describes the active ingredients specialized metabolites, phloroglu- 9 cinol-α-pyrone derivatives represent a structurally unique and pharmacologically signif- 10 icant class of compounds. This review consolidates over five decades of phytochemical 11 research, documenting 52 distinct compounds isolated from 11 Helichrysum species across 12 the Mediterranean, African, and Iranian regions.
Described compounds, have demonstrated meaningful bioactivity, including anti-inflammatory, antioxidant, antimicrobial, and neurobehavioural effects, suggesting their potential as molecular templates for drug development. This systematic synthesis of literature enhances the chemical understanding of Helichrysum phloroglucinol–α–pyrones and provides a comprehensive phytochemical reference. Beyond their pharmacological promise, these metabolites hold significance for chemotaxonomy, bioprospecting, and standardization of Helichrysum-based natural products.
The manuscript should be accepted for publication.
Author Response
Dear Reviewer. Sincere thanks for your positive feedback and your time to evaluate my mnauscript!
Reviewer 3 Report
Comments and Suggestions for Authors
Title and Abstract: It's not clear that this manuscript is a review. Must be clearly mentioned. What was the method of the review? The main conclusion is too general. What can be the most promising perspective?
It's not clear why the Helichrysum spp. are attractive and selected for the review. Please explain it.
Lines 49-51: Again, what was the method of data collection about the topic? Mention it shortly, please. Later, it is possible to read more.
Conclusions: The authors declare: "The genus Helichrysum remains one of the most chemically diverse representatives of 544 the Asteraceae family, remarkable for its vast array of specialized metabolites." What is the basis of the statement?
What kind of therapeutic potential do the authors mean here? The main conclusion is too general.
References: Only 30. Isn't there more? For a review article, it's not so much.
Author Response
Reviewer Comment 1: Title and Abstract: It's not clear that this manuscript is a review. Must be clearly mentioned. What was the method of the review? The main conclusion is too general. What can be the most promising perspective?
Answer Comment 1: Thank you for your valuable comment. The title has been changed to: "Phloroglucinol-α-Pyrones from Helichrysum Species: A Comprehensive Review of Isolation, Structural Diversity, and Biological Activities". The abstract and title mention that this manuscript is a review. The method of the review is mentioned in the Section "Literature Search Methodology".
Regarding the rationale for selecting Helichrysum: We focused on Helichrysum because it is the primary and most prolific source of phloroglucinol-α-pyrone derivatives, including arzanol—the most extensively studied compound in this class. Arzanol has attracted great scientific attention, it is derived from H. italicum and H. stoechas. By concentrating our review on Helichrysum, we were able to provide a systematic and comprehensive analysis of the chemistry and bioactivity of these specific compounds within a well-defined botanical context.
Based on the comprehensive bioactivity data for arzanol (the most studied compound in this natural compound class), the most promising perspectives for the described in the review substances are: (1) exploring the anti-inflammatory activity, especially towards dual mPGES-1/5-LOX inhibition that avoids COX-2 inhibitor cardiovascular toxicity, (2) exploring the potential brain glycogen phosphorylase activation for glycogen storage disorders and SIRT1 modulation for mood disorders, (3) cancer chemosensitizing through autophagy modulation profile and selective cytotoxicity, and (4) exploring the pharmacokinetics, especially for protein binding and solubility.
The method of the review is described in Section "Literature Search Methodology".
Reviewer Comment 2: "Lines 49-51: Again, what was the method of data collection about the topic? Mention it shortly, please. Later, it is possible to read more."
Answer Comment 2: Now in the Introduction section we have added a short paragraph detailing this.
Reviewer Comment 3: "Conclusions: The authors declare: "The genus Helichrysum remains one of the most chemically diverse representatives of 544 the Asteraceae family, remarkable for its vast array of specialized metabolites." What is the basis of the statement?
What kind of therapeutic potential do the authors mean here? The main conclusion is too general."
Answer Comment 3: Now the Conclusion section has been revised.
Reviewer Comment 4: "References: Only 30. Isn't there more? For a review article, it's not so much."
Answer Comment 4: We have conducted a thorough literature review dating from the first reports of isolated and characterized substances belonging to alpha pyrones - phloroglucinols, to present (late 2025). This collection represents all the literature that describes their extraction and subsequent structural elucidation.
We wish to thank again the Reviewer for his time and effort! Please accept our sincere gratitude!
Reviewer 4 Report
Comments and Suggestions for Authors
This is a very comprehensive article. Nevertheless, I still have some suggestions to offer.
1.I suggest modifying the format of Table 1 to make it look neater.
Author Response
Dear Reviewer, Thank you for your feedback. The structure of Table 1 organizes phytochemical data across multiple dimensions: plant species, compound classes, specific compounds with numerical identifiers, plant parts analyzed, and literature references.
The hierarchical indentation distinguishes between compound categories and individual compounds, making the information accessible. Given that this table consolidates findings from 20+ references across 13 Helichrysum species, with various compound types (dipyrones, prenyl phloroglucinols, benzofuranes, etc.) and plant parts, some visual density is inevitable and necessary to maintain comprehensive coverage within a single table.
Alternative formats (e.g., multiple smaller tables) would fragment the data and hinder comparative analysis across species. The current design balances completeness with readability for the specialized audience familiar with phytochemical nomenclature.